# Exploring Representations and Interventions in Time Series Foundation Models

**Michał Wiliński** [1]   **Mononito Goswami** [1]   **Willa Potosnak** [* 1]   **Nina Żukowska** [* 1]   **Artur Dubrawski** [1]

## Abstract

Time series foundation models (TSFMs) promise to be powerful tools for a wide range of applications. However, their internal representations and learned concepts are still not well understood. In this study, we investigate the structure and redundancy of representations across various TSFMs, examining the self-similarity of model layers within and across different model sizes. This analysis reveals block-like redundancy in the representations, which can be utilized for informed pruning to improve inference speed and efficiency. We also explore the concepts learned by these models, such as periodicity and trends. We demonstrate how conceptual priors can be derived from TSFM representations and leveraged to steer its outputs toward concept-informed predictions. Our work bridges representational analysis from language and vision models to TSFMs, offering new methods for building more computationally efficient and transparent TSFMs.

## 1. Introduction

Foundation models have taken significant strides in modeling both textual (Brown et al., 2020) and visual (Dosovitskiy et al., 2020) data, and have made complex language and image processing accessible to non-experts. These models are pretrained on massive internet-scale datasets and can be used to solve multiple tasks across a variety of domains, with little to no adaptation. Recently, a growing body of work (Garza & Mergenthaler-Canseco, 2023; Goswami et al., 2024; Rasul et al., 2024; Das et al., 2024; Woo et al., 2024; Ansari et al., 2024) has extended the benefits of this paradigm to time series data, a modality prevalent in fields such as finance (Taylor, 2008), healthcare (Goswami et al., 2021), and climate science (Schneider & Dickinson, 1974).

---
[*]Equal contribution [1]Auton Lab, School of Computer Science, Carnegie Mellon University. Correspondence to: Michał Wiliński <mwilinsk@andrew.cmu.edu>.

*Proceedings of the 42nd International Conference on Machine Learning*, Vancouver, Canada. PMLR 267, 2025. Copyright 2025 by the author(s).

Time series foundation models (TSFMs) have shown promising performance on multiple modeling tasks such as forecasting, classification, anomaly detection, and imputation, across a wide range of domains, and in settings with varying amounts of data and supervision. However, the underlying mechanisms and learned representations of TSFMs remain largely unexplored. The nature of learned representations in TSFMs remains largely unknown, including (1) their organizational structure, (2) the human-interpretable concepts they encode and their locations, and (3) how they can be manipulated to integrate prior knowledge for more informed predictions. Gaining deeper insight into the inner workings of TSFMs is key to improving their performance and trustworthiness.

We address each of these knowledge gaps by proposing approaches to systematically *analyze*, *probe*, and *intervene* in TSFMs through their learned representations. To address the first gap, we explore the fundamental question: **Do different TSFMs learn and organize knowledge in a similar manner?** This analysis is crucial for understanding how knowledge is organized in different models and whether representations are redundant across groups of layers, which can be potentially leveraged to improve inference efficiency. Next, we address the second gap by investigating: **What human-interpretable concepts can TSFMs learn, and where do these concepts emerge within the latent representations of the model?** Building on our layer-wise representation analysis, we propose new methods to identify and localize these concepts across layers. Finally, we tackle the third gap by asking: **Is it possible to intervene in TSFMs during inference by applying conceptual priors to steer time series predictions?** Through a novel approach, we demonstrate how to steer learned concepts by adjusting representations across layers, allowing the model to generate concept-informed predictions.

The main contributions of this paper are:

(i) **TSFM Representation Similarity Analysis and Pruning.** We translate prior work on vision transformers and convolutional neural networks to quantify the representation similarity in TSFMs. Our analysis reveals that model size substantially influences how representations are organized. In larger models, we observe high CKA similarity across groups of layers,

indicating redundant knowledge storage. Leveraging this insight, we propose a novel block-wise layer pruning strategy to remove redundant representations. This approach reduces model size and speeds up inference while maintaining accuracy.

(ii) **Concept Identification in TSFM Representations.** We introduce a new method to identify and localize specific time series concepts, such as trends and seasonal patterns, across model layers. Our findings show that TSFMs gradually learn these intuitive concepts across layers.

(iii) **Steering Methods for Concept-Informed TSFM Predictions.** We demonstrate a novel approach to steer model predictions by intervening in TSFM latent representations along specific conceptual directions (e.g., introducing an upward trend in forecasts) as a way to guide model output toward concept-informed predictions without additional training or fine-tuning.

## 2. Related Work

**Time Series Foundation Models (TSFMs).** TSFMs are versatile neural networks pretrained on vast amounts of time series data, and have shown impressive accuracy even in zero-shot settings. Several TSFMs have been proposed (Garza & Mergenthaler-Canseco, 2023; Liu et al., 2023; Das et al., 2024; Woo et al., 2024; Goswami et al., 2024; Ansari et al., 2024; Ekambaram et al., 2024; Talukder et al., 2024). Most of these are based on variations of the Transformer architecture (Vaswani et al., 2017), with differences in tokenization strategies, pretraining datasets, and the specific tasks they are designed to address. Consequently, we focus on developing methods to understand Transformer-based models. We specifically analyze three representative models: MOMENT (Goswami et al., 2024), Chronos (Ansari et al., 2024), and MOIRAI (Woo et al., 2024), which are fully open-source and offer distinct approaches to time series tokenization (patch vs. discrete), architecture (encoder-only vs. encoder-decoder), and pretraining objectives (imputation vs. forecasting). Recent advancements in TSFMs have focused mostly on improving their capabilities through scaling, specialization, or improving inference efficiency (Liu et al., 2024; 2023), as well as by incorporating long and multivariate contexts (Liu et al., 2025; Żukowska et al., 2024). However, there has been limited research on understanding what a specific TSFMs learn. For example, Goswami et al. (2024) showed that MOMENT learns some human interpretable concepts, while Ansari et al. (2024) investigated the failure modes of Chronos in controlled settings, and Potosnak et al. (2024) explored compositional reasoning within time series models, including TSFMs.

**Analyzing Representations of Deep Learning Models.**

Deep learning models often operate as black boxes, making their internal mechanisms and learned representations difficult to understand. One approach to gaining insights into these models is by comparing their intermediate representations. Similarity metrics can be used to determine the similarity or dissimilarity of representations at different stages of a model, revealing the hierarchy, homogeneity, and redundancy of learned features. Previous studies focused on quantifying the similarity of neural network representations (Raghu et al., 2017; Kornblith et al., 2019). (Raghu et al., 2021) demonstrated the use of these metrics for analyzing and comparing representations in vision transformers and CNNs, providing valuable insights into their functioning. (Nguyen et al., 2021) investigated the impact of model size and training data ratios on similarity patterns. They also explored model pruning based on similarity. Building on prior studies, we present the first comprehensive analysis of TSFM representations, providing insights into their internal mechanisms.

**Identifying Learned Concepts in Pretrained Models.** Understanding the internal representations learned by pretrained models has been an active area of research, particularly in the context of LLMs and vision models. Previous studies have explored whether individual neurons or directions in a model's latent space correspond to specific features or concepts (Dalvi et al., 2019; Goh et al., 2021; Gurnee et al., 2023; Elhage et al., 2022). These investigations often focus on identifying linear representations, where features are encoded as linear combinations of neuron activations. Recent work has also employed various probing techniques to classify and interpret these internal representations, addressing aspects such as truthfulness and model robustness (Azaria & Mitchell, 2023; Zou et al., 2023; Burns et al., 2023; Marks & Tegmark, 2024). While prior research has primarily focused on language and vision models, we demonstrate that simple techniques enable us to effectively probe TSFMs to identify and localize concepts.

## 3. Methods

We study TSFMs using three complementary methods of analysis and interventions concerning model layer-wise representations. In Section 3.1, we examine learned representations through the lens of similarity, uncovering stored knowledge redundancy in TSFM representations. We leverage this redundancy to prune multiple layers of pretrained models, thereby improving their efficiency. In Section 3.2, we identify specific concepts learned by TSFMs and localize them to specific model layers and token positions. In Section 3.3, we demonstrate how conceptual priors can be derived from TSFM representations and leveraged to steer its outputs toward concept-informed predictions.

## 3.1. Representation Similarity Analysis and Pruning

To gain a deeper understanding of TSFM representations, we investigate the following research questions: *(RQ1)* How similar are the representations learned by models of the same size but belonging to different families? *(RQ2)* How do these representations differ across models of varying sizes within the same family? *(RQ3)* How similar are the representations learned by corresponding layers of different TSFMs within the same family?

We consider several metrics commonly employed in the literature to analyze the similarity between learned representations. While our primary analysis relies on Centered Kernel Alignment (CKA) (Kornblith et al., 2019), we also explored Cosine Similarity and Singular Vector Canonical Correlation Analysis (SVCCA) (Raghu et al., 2017). For brevity, we provide a brief overview of CKA below, while detailed descriptions of the remaining metrics can be found in Appendix B.

**Representational Similarity using CKA.** CKA measures the similarity of representations by comparing the centered kernel matrices. It has been shown to be effective in capturing similarities between layers of deep networks (Kornblith et al., 2019). The general form of CKA between two sets of representations $\mathbf{X}$ and $\mathbf{Y}$ is defined as:

$$\text{CKA}(\mathbf{X}, \mathbf{Y}) = \frac{\text{HSIC}(\mathbf{X}, \mathbf{Y})}{\sqrt{\text{HSIC}(\mathbf{X}, \mathbf{X}) \cdot \text{HSIC}(\mathbf{Y}, \mathbf{Y})}} \quad (1)$$

where HSIC denotes the Hilbert-Schmidt Independence Criterion (Gretton et al., 2005).

For computational efficiency, we utilized a linear kernel in our CKA calculations, resulting in the following simplified formula:

$$\text{CKA}_{\text{linear}}(\mathbf{X}, \mathbf{Y}) = \frac{\|\mathbf{X}^T \mathbf{Y}\|_F^2}{\|\mathbf{X}^T \mathbf{X}\|_F \cdot \|\mathbf{Y}^T \mathbf{Y}\|_F} \quad (2)$$

where $\| \cdot \|_F$ denotes the Frobenius norm. The denominator in Equation 2 ensures that the metric value falls within the range of 0 to 1, facilitating interpretability. A high CKA value indicates a strong alignment between the two sets of representations, suggesting that the layers are likely learning similar features or concepts.

**Pruning TSFMs Based on Representational Similarity.** Large TSFMs typically learn redundant representations, which often manifest as block-like structures in heatmaps depicting pairwise similarity between layer activations (Figure 1). We leverage this redundancy to prune TSFMs, enhancing inference speed while preserving accuracy. Building on prior work (Nguyen et al., 2021), we propose a simple and effective layer pruning strategy, which we call *Block-wise Pruning*, outlined in Algorithm 1. To preserve the structural

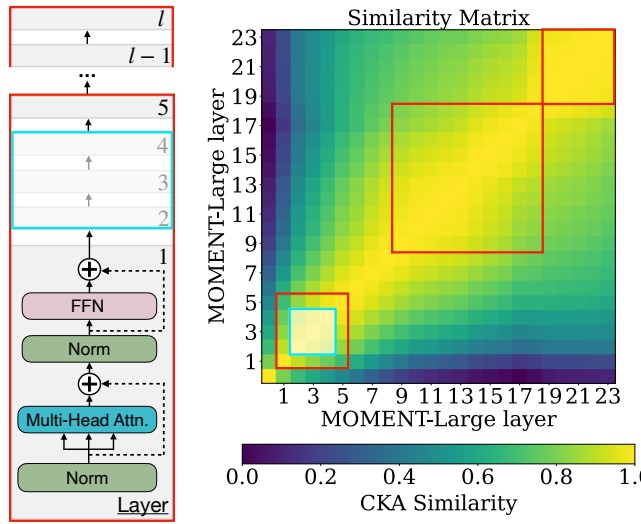

Figure 1: For each identified block of layers exhibiting redundant representations (red), we skip computation of their internal layers (blue). For example, if a block consists of five layers, we skip layers 2 through 4, retaining only the first and last layers to reduce representation redundancy while maintaining model integrity. More details on pruning can be found in Appendix C.

---

**Algorithm 1** Block-wise Pruning (Skipping Computation)

---

**Require:** Trained model $\mathcal{M}$ with layers $\{l_1, l_2, \ldots, l_n\}$;
   Identified redundant blocks $\mathcal{B} = \{b_1, b_2, \ldots, b_k\}$
   **for** each block $b_i$ in $\mathcal{B}$ **do**
      Let $b_i$ consist of layers $l_s$ to $l_e$ {Block edges at $l_s$ and $l_e$}
      **for** layer index $j = s + 1$ to $e - 1$ **do**
         Remove layer $l_j$ from model $\mathcal{M}$
      **end for**
   **end for**
   **return** pruned model $\mathcal{M}'$

---

integrity of each block, we retain the first and last layers of each block and skip computation by removing these blocks altogether.

To demonstrate the effectiveness of our proposed pruning strategy, we explore two pruning configurations, one in which we prune all redundant blocks, and the other where we prune only a single block. We compare the performance of these pruned models to the original, unpruned TSFMs using standard task-specific accuracy metrics (Mean Squared Error and Mean Absolute Error) and efficiency metrics (inference time in milliseconds and theoretical model size in megabytes). We evaluate these models on widely used imputation (Zhou et al., 2021) and forecasting (Ansari et al., 2024) benchmarks in both zero-shot settings and after linear

probing (Goswami et al., 2024). While prior work such as (Nguyen et al., 2021) primarily focused on pruning individual layers, our approach differs by targeting entire blocks of self-similar layers, preserving boundary layers to maintain representational continuity. Furthermore, we extend block-level pruning to large pretrained TSFMs, demonstrating practical effectiveness on diverse real-world tasks beyond classification, achieving substantial inference speedups (up to 52%) with minimal performance degradation, even in zero-shot scenarios.

### 3.2. Identifying and Localizing Time Series Concepts

Through our experiments, we aim to answer the following research questions: *(RQ4)* Do TSFMs represent concepts associated with specific data-generating functions distinctly in the latent space? *(RQ5)* Are these learned concepts localized to specific layers and tokens within TSFMs?

To systematically explore the ability of TSFMs to understand intuitive time series concepts, we randomly generate a large number of synthetic univariate time series. Each randomly generated time series belong to one of two pattern classes: *constant* or *sinusoidal*. Constant patterns, represented by $y(t) = mt + b$, capture long-term non-periodic trends. Sinusoidal patterns, modeled as $y(t) = a \sin\left(\frac{2\pi t}{f}\right)$, represent periodic processes. By controlling the parameters $m$, $b$, $a$, and $f$, we can systematically generate time series with varying slope, intercept, amplitude, and periodicity, respectively. Despite their simplicity, these data generation mechanisms capture a wide range of real-world time series patterns. Example series are shown in Fig. 3. For a detailed description of the data generation process, please refer to Appendix A.

**Identifying Linearly Represented Features.** We build on the investigation approach outlined in (Marks & Tegmark, 2024). We say that a feature is linearly represented in a foundation model $\mathcal{M}$ if it is represented as a *direction* in its latent space. If this feature is linearly represented in $\mathcal{M}$, we also want to identify which layer $l$ in $\mathcal{M}$ learns this concept in the most discriminant way. As a concrete example, consider that we want to check whether $\mathcal{M}$ can distinguish between constant and sinusoidal patterns.

To determine whether the feature (sinusoidal vs. constant time series) is linearly represented, we leverage the aforementioned synthetic dataset which comprises of multiple sinusoids and constant time series randomly sampled using our data generating function. Using this dataset, we extract the residual stream of each transformer block after the feed-forward layer. Let $\mathbf{h}_i^{(j)} \in \mathbb{R}^{n \times D}$ denote the hidden representation of a time series $\mathbf{x}$ at $i$-th layer and $j$-th token of $\mathcal{M}$, where $D$ is the dimensionality of the hidden layer. Linear probing involves training separate linear models for

each layer and token to classify time series $\mathbf{x}$ as a constant or sinusoid pattern. Classifiers $f_{ij}(\mathbf{h}_i^{(j)}, \theta_i^{(j)})$ are trained on the hidden representation $\mathbf{h}_i^{(j)}$ at each $i$-th layer and each $j$-th token to update the parameters $\theta_i^j$. Additionally, we perform probing on representations averaged along the token dimension for each $i$-th layer. The linear probes are trained to optimize the Fisher Criterion, a function that aims to maximize the distance between class means while minimizing within-class variance:

$$\mathcal{L}_{\text{Fisher}}(c, s) = -\frac{(\mu_s - \mu_c)^2}{\sigma_s^2 + \sigma_c^2}. \tag{3}$$

Here, $\mu_s$ and $\mu_c$ correspond to the mean embedding values, computed using all time series of a given class. Similarly, $\sigma_s^2$ and $\sigma_c^2$ correspond to the variance computed across the $n$ dimension for each class.

**Localizing Linearly Represented Features.** To localize which layers and tokens learn a specific concept, we compute the Fisher's Linear Discriminant Ratio (LDR) between the classes using the mean and variance statistics of $\mathbf{h}_i^{(j)}$ for each predicted class $\hat{\mathbf{y}}$, which is determined using the classifier $f_{ij}$ during linear probing. The goal of LDR is to maximize the separation between the classes by comparing the variance $\sigma^2$ within each class to the difference between the class means, $\mu$. A larger ratio indicates a clearer separation between the two classes, which can aid in concept localization by identifying where the classes are well-separated in the feature space. When applied in the context of neural network activations, LDR helps highlight which layers or features are most discriminative

$$\text{LDR}(\mathbf{h}_i^{(j)}|\hat{\mathbf{y}}) = \frac{(\mu_{\mathbf{h}_i^{(j)}|\hat{\mathbf{y}}=s} - \mu_{\mathbf{h}_i^{(j)}|\hat{\mathbf{y}}=c})^2}{\sigma_{\mathbf{h}_i^{(j)}|\hat{\mathbf{y}}=s}^2 + \sigma_{\mathbf{h}_i^{(j)}|\hat{\mathbf{y}}=c}^2} \tag{4}$$

$$= \frac{(\mu_s - \mu_c)^2}{\sigma_s^2 + \sigma_c^2}. \tag{5}$$

Here, $\mu_s$ and $\mu_c$ correspond to the mean computed across the $n$ dimension for each class. Similarly, $\sigma_s^2$ and $\sigma_c^2$ correspond to the variance computed across the sample dimension $n$ for each class. Let $\mathbf{V} = [v_{i,j}] \in \mathbb{R}^{L \times N}$ be the matrix of LDR values, where $v_{i,j}$ represents the LDR value for the $i$-th layer and $j$-th token, with $l$ layers and $N$ tokens. The LDR output is scaled between 0 and 1 using min-max scaling to allow for consistent comparison across layers.

By visualizing the scaled LDR values as shown in Figure 2, one can identify which layers and tokens exhibit the highest degree of separation between concept classes, offering insights into the network's internal representations for concept intervention techniques.

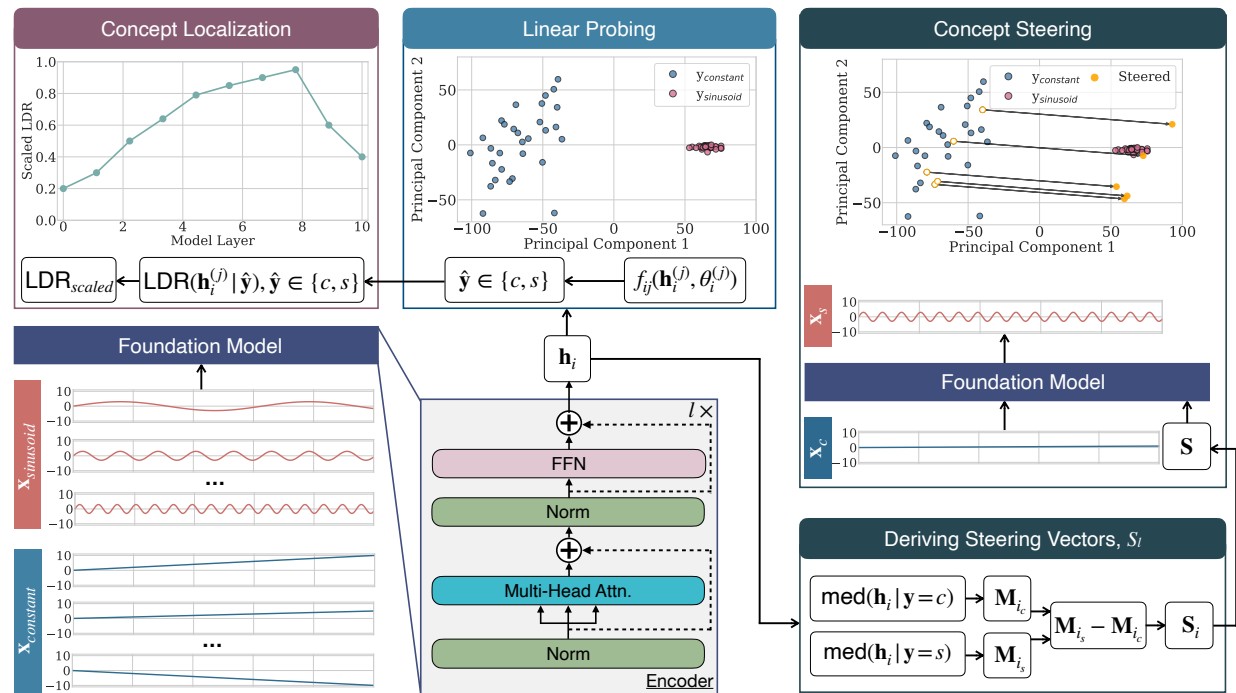

Figure 2: **Overview of linear probing, concept localization, and steering.** During linear probing we train linear classifiers $f_{ij}(\mathbf{h}_i^{(j)}, \theta_i^{(j)})$ to classify time series $\mathbf{x}$ into constant $c$ and sinusoid $s$ classes, using hidden representations $\mathbf{h}_i^{(j)}$ at the $i$-th layer and $j$-th token. We localize concepts using Fisher's Linear Discriminant Ratio (LDR) between the classes at each layer and token. The concept steering vector at the $i$-th layer is defined as the difference between the median activation matrices of the sinusoid and constant time series classes, $\mathbf{M}_{i_s} - \mathbf{M}_{i_c}$. Vectors of all layers are stacked into a steering matrix $\mathbf{S}$ to steer model predictions towards desired concepts by updating the embeddings as $\mathbf{h}_i \leftarrow \mathbf{h}_i + \lambda \mathbf{S}_i$, where $\lambda$ is a scalar that controls the strength of the intervention.

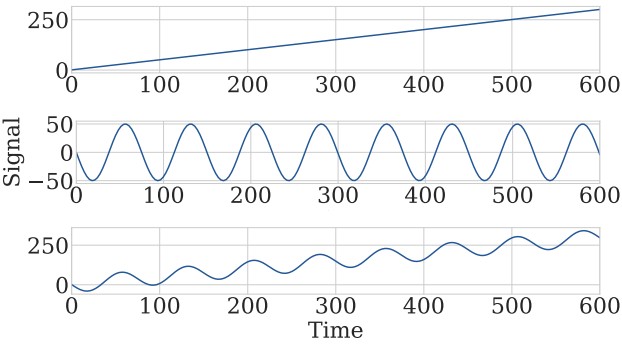

Figure 3: Examples of synthetic data generated for concept localization and steering experiments include signals with varying trends (top), sinusoidal signals with varying frequencies (middle), and combinations of signals with varying trends and sinusoidal patterns (bottom).

### 3.3. Concept-Informed Predictions via Model Steering

Through our experiments, we aim to answer the following research questions: *(RQ6)* Can we leverage these learned concepts to steer model output toward concept-informed

predictions? For example, can we add periodicity or an upward trend to a constant time series? *(RQ7)* Is it possible to combine multiple steering interventions to manipulate model predictions towards complex compositions of various concepts? For instance, can we steer a model to add both trend and periodicity to a constant signal?

**Deriving Steering Matrices for Model Steering.** Once we have identified that a feature is linearly represented in the latent space of the $\mathcal{M}$, we can use steering interventions to manipulate the latent space and generate time series that reflect intended concepts. For instance, to introduce periodicity to a constant time series, we can utilize a steering matrix $\mathbf{S}$, as illustrated in Figure 2. By strategically intervening in $\mathcal{M}$ using this steering matrix, we can bias its outputs towards predicting periodic time series. To construct a steering matrix, we first derive steering vectors $\mathbf{S}_i \in \mathbb{R}^{N \times D}$, for each layer $i$. These vectors represent the change that activations in layer $i$ must undergo such that $\mathcal{M}$ produces periodic outputs. $\mathbf{S}_i$ is simply the difference between the *median* activation matrix of the constant time series $\mathbf{M}_{i_c}$, from that of sinusoids $\mathbf{M}_{i_s}$. We stack these vectors for all layers to derive the steering matrix. This matrix allows us to

simultaneously intervene across multiple tokens and layers during inference, which we found to be more effective than single-token interventions. During inference, to steer model predictions, at each layer $i$, we update its hidden representation as follows: $\mathbf{h}_i \leftarrow \mathbf{h}_i + \lambda \mathbf{S}_i$, where $\lambda \in \mathbb{R}$ is a scalar that controls the strength of the intervention.

We explore two intervention techniques: (1) deriving steering vectors using the mean of hidden activations rather than the median, and (2) steering a single token versus all tokens throughout the model. While our methods are applicable to a wide range of transformer-based foundation models, we focus on two prominent TSFM families for brevity: MOMENT (Goswami et al., 2024) and Chronos (Ansari et al., 2024). Both these models are fully open-source, come in different sizes, yet have fundamentally different design choices. For example, Chronos is based on encoder-decoder transformer (in our work we will investigate encoder part) model which takes discretized time series as input, whereas MOMENT is a multi-task, encoder-only model which takes continuous time series patches as input. We supplement our representation analysis results with Moirai (Woo et al., 2024), another popular TSFM which comes in different sizes. More information on hyperparameters can be found in Appendix E.

## 4. Results

**Analyzing representations offers interesting insights.** Our analysis of model representations demonstrates that both model size and internal architecture considerably influence how representations are organized. Fig. 7 shows heatmaps which reveal that larger models, such as MOMENT-Large, Chronos-Large, and Moirai-1.1-R Large, have similar representations across specific groups of layers forming distinct and intricate block patterns, which may reflect unique stages of representation learning. More complex block patterns are observed with increasing model size, indicating that scaling may enhance the richness and organization of internal representations. However, it may also increase redundant knowledge storage through similar representations across layers, as suggested by high CKA similarity measured in block patterns. Interestingly, within model families (e.g., Chronos and Moirai), scaling does not always result in predictable heatmap changes. Larger models, like Chronos-Large and Moirai-Large, demonstrate more refined and complex transformations of representations that are not easily extrapolated from their smaller versions as shown in Fig. 15). Moreover, cross-model similarity analysis results in Fig. 4 reveal that while early layers tend to have high similarity across models of different sizes, the similarity measures among later layers diverge more notably, particularly in larger models. This divergence is especially evident in the

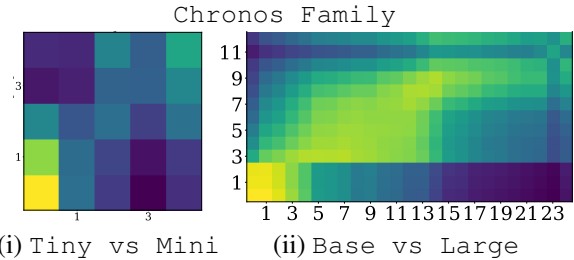

Chronos Family

(i) Tiny vs Mini        (ii) Base vs Large

Figure 4: Similarity between representations learned by different layers in TSFMs of the same family but different sizes. Initial layers tend to learn similar representations, while the similarity gradually decreases in the later layers.

Chronos family, where early representations are more consistent across models, but later layers become increasingly specialized as model depth increases as shown in Fig. 6.

**Block-wise pruning can improve model throughput, without compromising accuracy.** We observed consistent improvements in memory efficiency and inference speed compared to unpruned model counterparts. For example, pruning only Block 3 for MOMENT-Large, resulted in a 11% decrease in estimated model size with a 5% decrease in inference time. Furthermore, this pruned model had lower zero-shot imputation MAE for 5 of 9 datasets (ETTh2, ETTm1, ETTm2, Exchange, and Weather) as shown in Tab. 3. Chronos-Large results for zero-shot experiments are reported in Tab. 4. Detailed results on memory usage and speed improvements can be found in Tab. 7. Although pruning consistently improved memory efficiency and inference speed compared to unpruned counterparts, performance varied between pruning methods and datasets, and some methods showed considerable degradation. In addition to zero-shot experiments, we conducted fine-tuning experiments on pruned models—including MOMENT-Large with all block redundancies removed—to evaluate the impact of the most aggressive pruning approach on forecasting performance. Notably, the pruned model performed on par with its unpruned baseline, as shown in Table 1, demonstrating the potential of our block-wise pruning approach to maintain performance while reducing model complexity. Complete finetuning results are provided in Table 6 in Appendix G.

**TSFMs learn intuitive linear concepts.** Our concept localization results in Fig. 7 show that certain concepts represented by MOMENT-Large are linearly separable and that this separability is not consistent but rather emerges at specific layers in the model. We also found intuitive differences in the locations where these concepts are learned. We observed that certain concepts, such as distinguishing between constant and sinusoidal patterns, require examination of the entire time series. In contrast, differentiating between in-

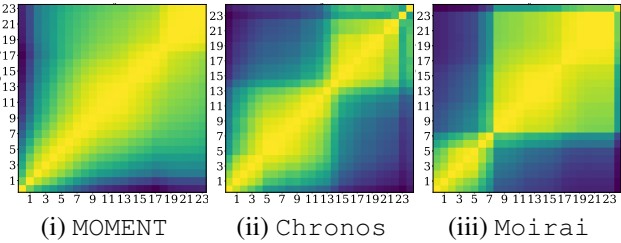

| (i) MOMENT | (ii) Chronos | (iii) Moirai |

Figure 5: Pairwise similarity of layer measured using CKA for large variants of TSFMs (dark blue → low similarity, yellow → high similarity). All TSFMs learn redundant representations with manifesting as block-like structures.

| | ETTh1 | ETTh2 | ETTm1 | ETTm2 | ILI | Weather |
|---|---|---|---|---|---|---|
| Vanilla | **0.385** | **0.287** | **0.290** | **0.171** | 3.260 | 0.153 |
| Pruned | 0.388 | 0.296 | **0.290** | 0.173 | **2.981** | **0.152** |

Table 1: MSE of fine-tuned vanilla and pruned MOMENT variants on long-horizon forecasting datasets (Zhou et al., 2021). Pruned models perform on par with the original, while reducing memory consumption by > 50% and improving inference time per sample by ≈1 ms. Full results are provided in Tables 6 and 7 in the Appendix.

Chronos Family

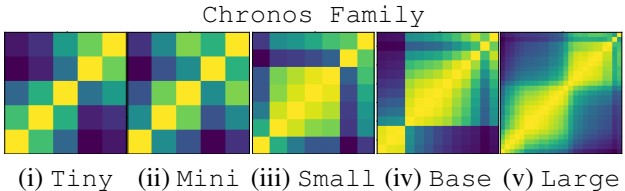

(i) Tiny  (ii) Mini (iii) Small (iv) Base (v) Large

Figure 6: Does model size influence the patterns of learned representations? The emergence of block-like patterns in larger models cannot be directly inferred from the patterns observed in smaller models. (expanded view is Fig. 16).

creasing and decreasing trends can be achieved by focusing on the initial and final patches. However, we did not identify specific locations where models learn to distinguish between time series of different amplitudes. This may be attributed to the normalization of input time series, a common practice in many TSFMs, including MOMENT.

**We can effectively steer TSFM predictions.** Our concept steering interventions effectively transform the latent space of TSFMs, resulting in model predictions that align with the intended concepts, as demonstrated in Fig. 8. We successfully introduced periodicity and trend concepts to constant time series and demonstrated the ability to combine multiple steering vectors to create more complex patterns. By combining steering vectors representing increasing trends and sinusoidal patterns, we were able to steer model predictions towards a combination of these features. To assess

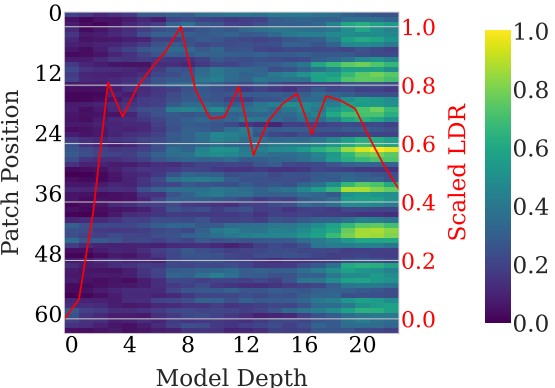

Figure 7: Linear separability of concepts at the patch level (y-axis) across model layers (x-axis). Yellow indicates higher separability than blue in the colorbar. Linear separability is measured using the Linear Discriminant Ratio (LDR), derived from model embedding statistics for each predicted class (constant vs. sinusoidal patterns). Certain concepts captured by MOMENT-Large are linearly separable, but this separability is not uniform; it emerges at specific layers and patches of the model. More examples of linear separability of concepts are shown in Fig. 14.

the robustness of concept steering, we generated datasets using multiple random seeds and confirmed that consistent steering effects emerge across samples. This is supported by analyses of linear concept separability and concept emergence in the latent space. Furthermore, in Appendix F, we present steering results on real-world ECG data, demonstrating how to steer time series from the normal to the abnormal heartbeat class in the ECG5000 dataset. Our concept steering method can be effectively leveraged on both synthetic and real-world data, helping to validate its robustness and practical applicability.

To assess the impact of steering in the latent space, we analyzed changes in the hidden representations before and after applying concept steering by projecting them into a two-dimensional space using Principal Component Analysis (PCA). We found that steering in the latent space is reflected in these lower-dimensional representations, as illustrated in Fig. 11. The PCA reduction often captured the concept direction as one of the principal components. This could be attributed to the distinct concepts in our synthetic data.

Interestingly, the method of obtaining the steering matrix, either by computing the mean or median across embedding concept classes, has no notable effect on the steered output as shown in Fig. 12. However, applying concept steering interventions across all tokens is necessary to achieve the intended steered concept output compared to applying concept steering interventions to a single token. Moreover, the $\lambda$ parameter can have considerable effect on steered output. For

**Concept Steering:** *Use conceptual priors to shape time series predictions.*

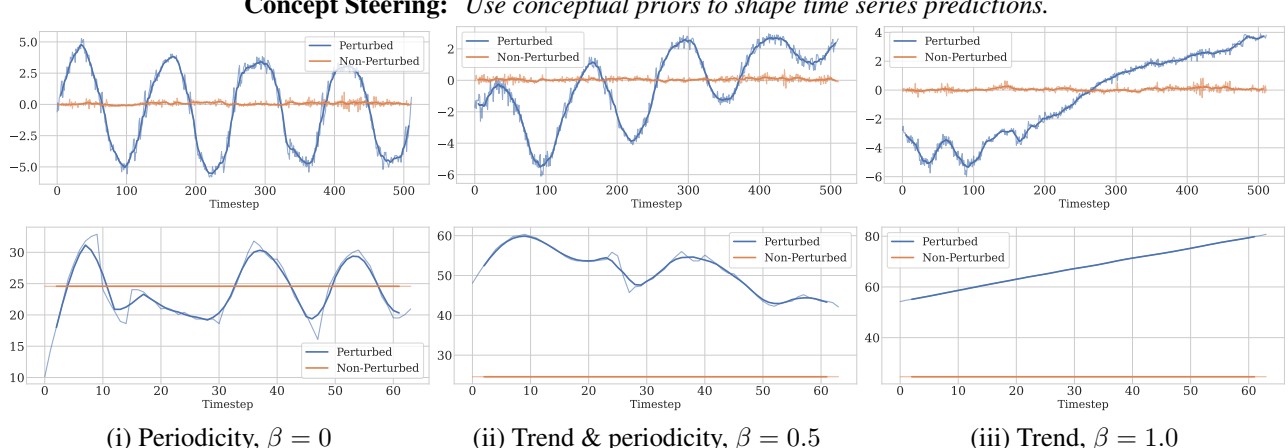

(i) Periodicity, $\beta = 0$   (ii) Trend & periodicity, $\beta = 0.5$   (iii) Trend, $\beta = 1.0$

Figure 8: Visualization of `MOMENT` reconstruction and `Chronos` forecasting predictions (bottom), comparing concept steering in the latent space (blue) with the unsteered baseline (orange). The original constant signal and steered output are referred to as Non-perturbed and Perturbed in the figure legend, respectively. A constant time series is first provided as input to both the proposed steering approach and the unsteered baseline. The non-perturbed output remains a constant signal, as expected, while the perturbed output reflects the new concept introduced via the steering matrix—such as trend, seasonality, or both—depending on the specified $\beta$ parameter value. For both outputs, we show the raw model predictions (lighter color) and their moving averages (darker color) to reduce noise artifacts. Steering results are shown for the following experiments: (i) steering a constant signal to have periodicity ($\beta = 0$), (ii) steering a constant signal to have an increasing slope, or trend ($\beta = 1.0$), and (iii) steering a constant signal to have trend and periodicity ($\beta = 0.5$).

`Chronos`, steering required tuning the parameter $\lambda \approx 0.1$ for effective performance, whereas `MOMENT` maintained effective steering with $\lambda = 1$.

## 5. Discussion

We explored two complementary approaches to probing and intervening in TSFMs. We gained valuable insights into their internal mechanisms and identified opportunities for improvement. For instance, our analysis revealed redundancy in learned representations. We leveraged this representational redundancy, inherent to large over-parameterized TSFMs, to devise a simple block-wise pruning strategy. This strategy effectively reduced the size and computational cost of these models without compromising performance, demonstrating the potential for distilling smaller, more efficient models from larger TSFMs.

We also explored ways to use conceptual priors to shape time series predictions. Concept steering enables controlled, post-training updates to model embeddings through a lightweight steering matrix, allowing models to incorporate new concepts or events into predictions without requiring retraining or fine-tuning. In addition to reducing the computational cost of further training, concept steering has many practical applications. It enables users to imbue models with new or missing contextual factors after training, which can be particularly valuable if models do not include certain scenarios in

their original training. For example, we can steer vital signs in healthcare based on new treatments or adjust financial forecasts based on emerging events such as positive earnings surprises. Adding these contextual factors, even as simple trends, has significant implications for improving model predictions in zero-shot and out-of-distribution scenarios. Steering techniques can also support realistic synthetic generation of time series variations. In our experiments with the ECG Arrhythmia classification dataset (ECG5000), we demonstrate that time series classified as normal heartbeats can be steered to produce time series classified as abnormal heartbeats. Such data generation capabilities can be used to augment data for model training or generate new samples for us to better understand the decision boundaries in model predictions. Prior work has already shown that data augmentation techniques can improve TSFM generalization by enhancing model robustness and exposing it to a wider variety of patterns (Ansari et al., 2024).

**Limitations and Future Work.** While our methods are broadly applicable to different transformer-based foundation models, future research should explore other architectures, such as state space models (Gu & Dao, 2024) and stacked multi-layer perceptrons (Ekambaram et al., 2024). Moreover, future studies could evaluate steering methods on other real-world time series datasets and domains, as well as on tasks beyond forecasting and imputation, such as anomaly detection and classification.

## Impact Statement

While our pruned TSFMs demonstrate promising performance, it is crucial to exercise caution when using them, especially in high-stakes applications such as healthcare. Before deploying these models for critical decision-making, we strongly recommend fine-tuning and evaluating them on task-specific, in-domain datasets using relevant metrics. The ability to steer model predictions offers numerous benefits but also raises concerns regarding potential biases. We urge users to exercise caution when using our proposed steering strategies and to carefully consider the potential implications of manipulating model outputs.

## Reproducibility Statement

To ensure reproducibility, we have made our code anonymously accessible through `GitHub`. All time series foundation models used for our analysis are publicly available and open-source: `Chronos`, `MOMENT`, and `MOIRAI`. All models were trained and evaluated on a computing cluster consisting of 128 AMD EPYC 7502 CPUs, 503 GB of RAM, and 8 NVIDIA RTX A6000 GPUs, each with 49 GiB RAM.

## Acknowledgment

**Funding** This work was partially supported by NSF (awards 2406231 and 2427948), NIH (awards R01NS124642, R01DK131586, and R01NR013912), and the US Army (W911NF-20-D0002).

**Datasets and Tooling** We are grateful to the creators and maintainers of the UCR Time-Series Archives (classification & anomaly) (Dau et al., 2018; Keogh et al., 2021), the Monash Forecasting Archive team (Godahewa et al., 2021), the Informer long-horizon data providers (ETT, Electricity, Exchange Rate, Weather, Traffic, ILI) (Zhou et al., 2021), the custodians of TSB-UAD (Paparrizos et al., 2022), and every other public dataset we relied on. We would also like to thank the open-source community– authors of the libraries, frameworks, and tooling that make modern machine learning research possible.

**Discussions** We thank the anonymous reviewers and meta-reviewer for their constructive feedback and suggestions, which helped improve the quality and clarity of this work. We are grateful to Carnegie Mellon University (CMU), the Robotics Institute, and the Auton Lab for providing the research environment and infrastructure that made this work possible. We also thank Rachel Burcin and Dr. John Dolan for their support during the CMU Robotics Institute Summer Scholars program. Additionally, we thank our fellow Auton Lab members Jieshi Chen, Piotr Bartosiewicz, Chi-En Teh, Nicholas Gisolfi, Ignacy Stepka, and Kacper Trebacz for inspiring discussions and support that helped shape this work.

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

# A. Synthetic Data Generation

Examples of synthetic data are provided in Fig. 9. Generated time series $\{x_t\}_{t=1}^T$ patterns include:

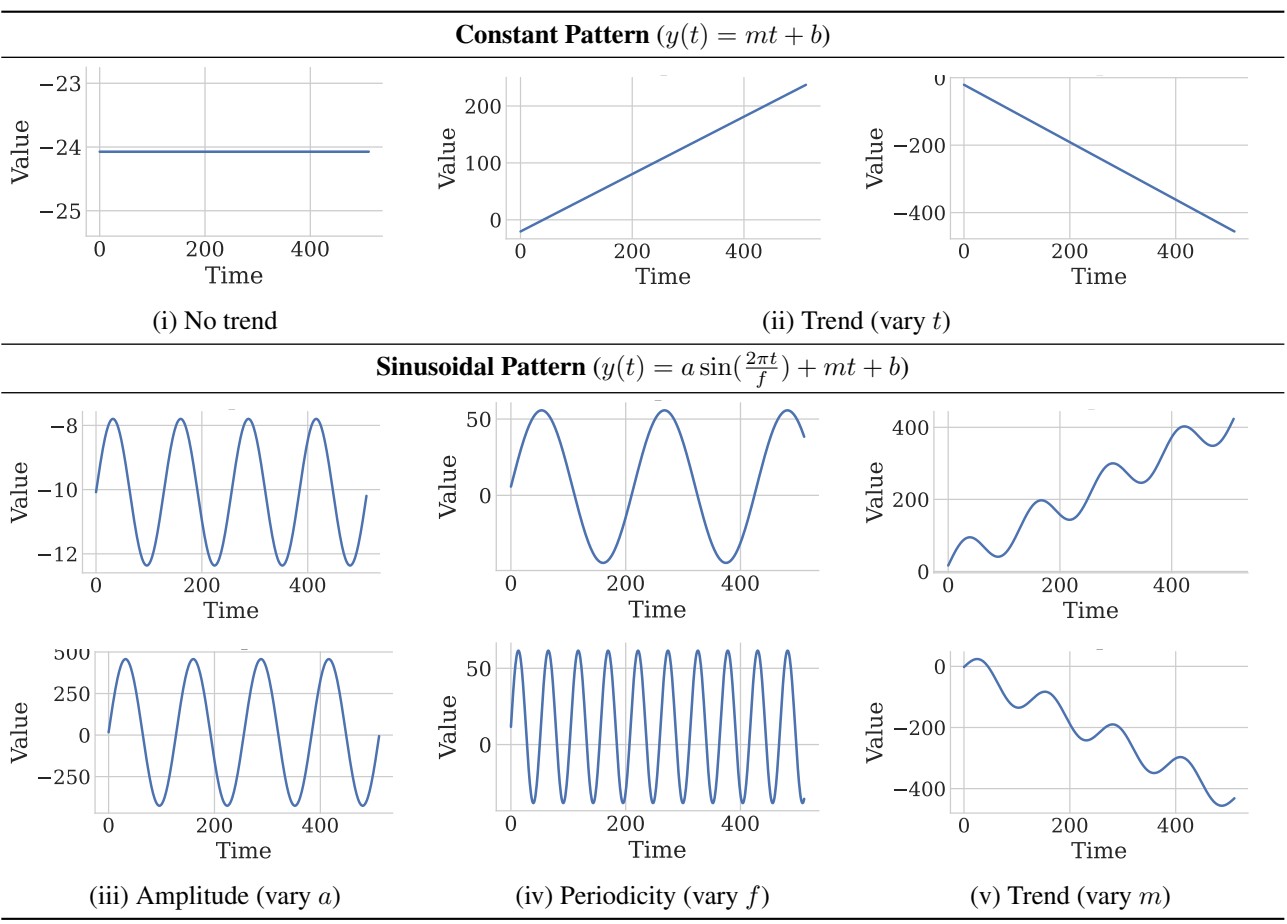

Figure 9: Samples of synthetic time series used in our experiments. We use two base patterns (constant and sinusoidal). To generate synthetic datasets, we vary the periodicity $f$, amplitude $a$, intercept $b$, and linear trend $m$.

- **Constant Pattern.** The constant pattern captures long-term progression of the time series and is modeled as:

$$y(t) = mt + b,$$

  where $\alpha$ is the slope and $\beta$ is the intercept. These parameters are sampled from uniform distributions:

$$m \sim U(\text{slope}_{\text{min}}, \text{slope}_{\text{max}}),$$

$$b \sim U(\text{intercept}_{\text{min}}, \text{intercept}_{\text{max}}).$$

- **Sinusoidal Pattern.** The sinusoidal pattern captures patterns of periodic variations and is modeled as:

$$y(t) = a\sin\left(\frac{2\pi t}{f}\right) + mt + b,$$

  where $A$ is the amplitude and $P$ is the period, both sampled from uniform distributions:

$$a \sim U(\text{amplitude}_{\text{min}}, \text{amplitude}_{\text{max}}),$$

$$f \sim U(\text{period}_{\text{min}}, \text{period}_{\text{max}}).$$

Parameter Ranges:

- **Constant Case:**
  $a \sim U(0,0), \quad f \sim U(0,0), \quad m \sim U(0,0), \quad b \sim U(-30, 30).$

- **Increasing Slope Case:**
  $a \sim U(0,0), \quad f \sim U(0,0), \quad m \sim U(0.5, 1), \quad b \sim U(-30, 30).$

- **Decreasing Slope Case:**
  $a \sim U(0,0), \quad f \sim U(0,0), \quad m \sim U(-1, -0.5), \quad b \sim U(-30, 30).$

- **Sine Constant Case (with seasonality parameters):**
  $a \sim U(50, 50), \quad f \sim U(128, 128), \quad m \sim U(0,0), \quad b \sim U(-30, 30).$

- **Sine Increasing Slope Case (with seasonality parameters):**
  $a \sim U(50, 50), \quad f \sim U(128, 128), \quad m \sim U(0.5, 1), \quad b \sim U(-30, 30).$

- **Sine Decreasing Slope Case (with seasonality parameters):**
  $a \sim U(50, 50), \quad f \sim U(128, 128), \quad m \sim U(-1, -0.5), \quad b \sim U(-30, 30).$

## B. Additional Representation Similarity Metrics

To comprehensively analyze the similarity of learned representations, we considered several metrics commonly used in the literature. While our primary analysis relies on Centered Kernel Alignment (CKA) (Kornblith et al., 2019), we also explored two additional similarity metrics, namely Cosine Similarity and Singular Vector Canonical Correlation Analysis (SVCCA) (Raghu et al., 2017). Our findings consistently demonstrated similar patterns across all these metrics, underscoring the robustness of our results. We provide brief descriptions of Cosine Similarity and SVCCA below.

**Cosine Similarity:** Cosine similarity measures the cosine of the angle between two vectors, providing a simple yet effective way to assess similarity. In our case, we work with activation matrices for multiple samples and compute the average cosine similarity. Given two matrices $\mathbf{X}$ and $\mathbf{Y}$, representing the activations from two layers, the cosine similarity is computed as:

$$\text{cosine similarity}(\mathbf{X}, \mathbf{Y}) = \frac{1}{n} \sum_{i=1}^{n} \frac{\mathbf{x}_i \cdot \mathbf{y}_i}{\|\mathbf{x}_i\| \|\mathbf{y}_i\|} \tag{6}$$

where $\mathbf{x}_i$ and $\mathbf{y}_i$ are the $i$-th columns of matrices $\mathbf{X}$ and $\mathbf{Y}$, and $n$ is the number of samples.

**Singular Vector Canonical Correlation Analysis (SVCCA) (Raghu et al., 2017):** SVCCA is a method that compares the similarity of representations by aligning subspaces spanned by the top singular vectors. It effectively reduces the dimensionality and then compares the correlations of the principal components. The SVCCA similarity between two activation matrices $\mathbf{X}$ and $\mathbf{Y}$ is computed as follows:

$$\text{SVCCA}(\mathbf{X}, \mathbf{Y}) = \text{CCA}(\mathbf{U}_k, \mathbf{V}_k) \tag{7}$$

where $\mathbf{U}_k$ and $\mathbf{V}_k$ are the top $k$ singular vectors obtained from the singular value decomposition (SVD) of $\mathbf{X}$ and $\mathbf{Y}$, respectively, and CCA denotes the canonical correlation analysis.

## C. Finding and Pruning Redundant Blocks in TSFMs

The term "block" refers to groups of consecutive layers within a transformer that exhibit high representational similarity. Consistent with prior work (Phang et al., 2021), we use "layer" to refer to individual transformer encoder or decoder blocks consistent with prior work, while "block" refers to a higher-level structure made up of multiple such layers that share similar representations (Nguyen et al., 2021).

As an example, consider Figure 1 which illustrates the pairwise similarity between the 24 layers of `MOMENT-Large`. In this figure, lighter colors (yellow) represent higher representational similarity, as measured by Centered Kernel Alignment (CKA) (Kornblith et al., 2019). The identified blocks are outlined with red bounding boxes. For example, Block 1 comprises layers 1–5, Block 2 comprises layers 9–18, and Block 3 comprises layers 19–23.

Our pruning algorithm is summarized in Algorithm 1. Our pruning method involves retaining the first and last layers of each block while zeroing out the weights of the intermediate layers. This approach preserves the structural integrity of the block while leveraging the skip connections within transformer blocks to ensure that signals and gradients continue to flow through the network. In the case of `MOMENT-Large`'s Block 3, composed of layers 19–23, this means that layers 19 and 23 are retained, while the weights of layers 20, 21, and 22 are skipped. We have added a dedicated section on pruning in Appendix C to further clarify this process.

---

**Algorithm 2** Block Identification and Filtering

---

**Require:** CKA similarity matrix $S$, similarity threshold $\tau$, minimum block size $k$
  Initialize blocks $\leftarrow \emptyset$
  Initialize current_block $\leftarrow \emptyset$
  **Phase 1: Initial block identification**
  **for** each encoder block $l_i$ **do**
    **if** $\forall l_j \in$ current_block : $\text{CKA}(l_i, l_j) \geq \tau$ **then**
      current_block $\leftarrow$ current_block $\cup \{l_i\}$
    **else**
      **if** $|$current_block$| \geq k$ **then**
        blocks $\leftarrow$ blocks $\cup \{$current_block$\}$
      **end if**
      current_block $\leftarrow \{l_i\}$
    **end if**
  **end for**
  **Phase 2: Filter small blocks**
  filtered_blocks $\leftarrow \{b \in$ blocks $: |b| \geq k\}$
  **Phase 3: Verify block-wide self-similarity**
  Initialize final_blocks $\leftarrow \emptyset$
  **for** each block $b \in$ filtered_blocks **do**
    Let start, end be the indices of first and last layer in $b$
    submatrix $\leftarrow S[$start : end, start : end$]$
    min_similarity $\leftarrow \min($submatrix$)$
    **if** min_similarity $\geq \tau$ **then**
      final_blocks $\leftarrow$ final_blocks $\cup \{b\}$
    **end if**
  **end for**
  **return** final_blocks

---

### C.1. A Simple Algorithmic Approach to Identify Redundant Blocks

We identify redundant blocks in a TSFM through visual inspection, which aligns with prior work (Nguyen et al., 2021). Table 2 lists all the identified blocks in `MOMENT-Large` and `Chronos-Large`. In addition to visual inspection, redundant blocks can also be identified using algorithmic approaches.

Below we propose a simple algorithm to identify redundant blocks in TSFMs. First, we can define a block as:

$$\text{Block} = \{l_i, ..., l_j\}$$

where $i < j$ and $l_k$ are adjacent transformer encoder blocks

Then, we can systematically identify these blocks using an algorithm that:

1. Identifies initial candidate blocks by grouping adjacent layers with pairwise CKA similarity above a threshold

2. Filters out blocks that are too small to be considered meaningful

3. Verifies that each block exhibits high similarity across all its constituent layers by examining the complete submatrix of similarities

### C.2. Identified Blocks in `MOMENT` & `Chronos`

In this paper, identified redundant blocks visually. Below we specify, the redundant blocks for `MOMENT-Large` and `Chronos-Large`:

| Models | Block 1 | Block 2 | Block 3 | Block 4 |
|---|---|---|---|---|
| `MOMENT-Large` | $1-5$ | $9-18$ | $19-23$ | N/A |
| `Chronos-Large` | $1-4$ | $5-9$ | $10-13$ | $15-22$ |

Table 2: Visually identified blocks of redundant layers in `MOMENT-Large` and `Chronos-Large`.

## D. On Steering and the parameter $\lambda$

Below we provide some guidance on selecting good values of $\lambda$ and insights into its properties:

**Selection and Impact**

- **Optimal Range:** Based on our empirical experiments, we found that the steering strength parameter $\lambda$ is most effective for interventions when its value lies within the interval $[0.1, 2.0]$.

- **Lower Bound Considerations:** Values of $\lambda < 0.1$ often result in insufficient perturbation of the activation patterns, leading to suboptimal intervention effects that may not manifest visibly in the model's output.

- **Upper Bound Effects:** Setting $\lambda > 2.0$ induces excessive perturbations that push activations beyond their typical distribution bounds, potentially resulting in degenerate or semantically meaningless outputs. In the PCA/latent space visualizations, these cases simply appear as more distant points along the steering direction.

**Directional Properties**

- **Reversibility**: Multiplying the steering vector by -1 effectively reverses the direction of intervention, enabling bidirectional control (e.g., transforming concept $A \to B$ into $B \to A$).

- **Example application**: For a steering vector trained to increase signal magnitude, applying its negative counterpart $(-\lambda \mathbf{S})$ produces controlled signal decrease, demonstrating the symmetric nature of the steering operation.

**Practical Guidelines**

- **Initial Calibration**: We recommend starting with $\lambda = 1.0$ and adjusting based on the observed intervention strength. In most cases value $\lambda = 1.0$ works well and does not need tuning.

- **Task and Model-Specific Tuning**: If $\lambda = 1.0$ does not yield satisfactory results, the optimal value requires tuning based on both the specific steering objective and target model, necessitating empirical calibration to achieve the desired intervention strength.

- **Monitoring**: When applying steering interventions, practitioners should monitor both the immediate output and latent space representations to ensure meaningful transformations while maintaining output coherence.

## E. Model Descriptions and Specifications

### E.1. MOMENT

The MOMENT model family is designed for general-purpose time-series analysis, utilizing an **encoder-only** Transformer architecture. It processes input time series by dividing them into fixed-length sub-sequences (patches) and encoding each patch into a D-dimensional space. The pre-training task involves reconstructing masked patches to learn robust representations that generalize across various tasks.

- **Architecture**: Encoder-only Transformer, with patch embeddings and masking for reconstruction.

- **Input Representation**: Time series split into fixed-length patches, embedded into D-dimensional vectors.

- **Model Variants**: MOMENT-Large

### E.2. Chronos

Chronos models utilize a **sequence-to-sequence (encoder-decoder)** Transformer architecture based on the T5 model family. Time series are scaled and quantized into tokens, which are processed by the encoder. The decoder autoregressively predicts future time steps, generating tokens that are mapped back into numerical values.

- **Architecture**: Encoder-decoder Transformer based on T5, with a reduced vocabulary size of 4096 tokens.

- **Input Representation**: Time series quantized into discrete tokens for sequence modeling.

- **Model Variants**: Chronos is available in the following configurations:
    - `chronos-t5-tiny`: 8M parameters
    - `chronos-t5-mini`: 20M parameters
    - `chronos-t5-small`: 46M parameters
    - `chronos-t5-base`: 200M parameters
    - `chronos-t5-large`: 710M parameters

### E.3. Moirai

Moirai is a time series foundation model built with an **encoder-only** Transformer architecture designed for universal forecasting. The model handles varying temporal resolutions with multiple patch size projection layers and uses any-variate attention for multivariate time series. A mixture distribution is employed to model probabilistic forecasts.

- **Architecture**: Encoder-only Transformer with any-variate attention for multivariate time series forecasting.

- **Input Representation**: Time series processed using multi-patch size projections to handle different frequencies.

- **Model Variants**: Moirai is available in three configurations:
    - `Moirai-small`: 14M parameters
    - `Moirai-base`: 91M parameters
    - `Moirai-large`: 311M parameters

## F. Additional Steering Results

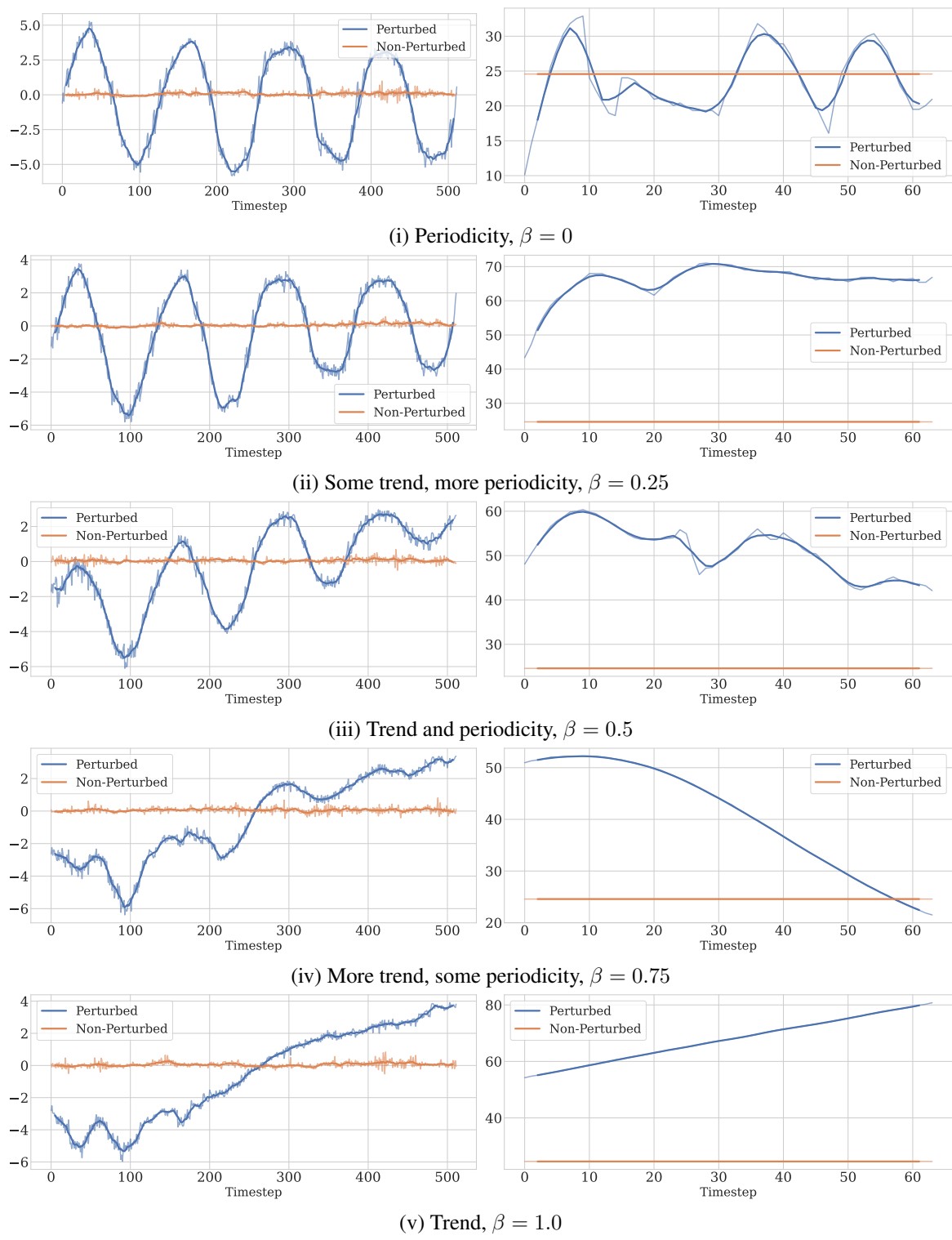

Figure 10: Visualization of the `MOMENT` model's reconstruction (left) and the `Chronos` model's forecasting (right) predictions, comparing concept steering in the latent space (blue) with the unsteered baseline (orange). The steering parameter $\beta$ can be used to interpolate between sinusodial concepts to increasing trend concepts by varying $\beta \in [0.0, 1.0]$.

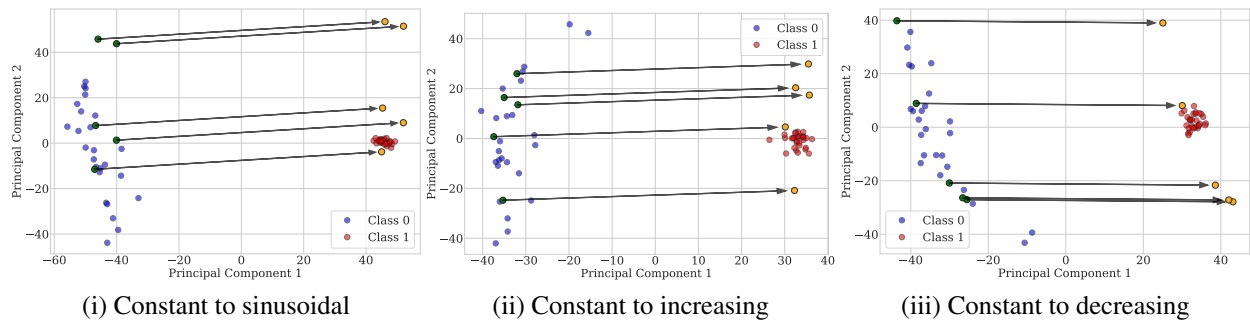

(i) Constant to sinusoidal      (ii) Constant to increasing      (iii) Constant to decreasing

Figure 11: Visualization of steering effects in latent space reduced using PCA analysis. Here, we considered steering constant to sinusoidal, constant to increasing, and constant to decreasing series. As shown, the steering behaves as expected in the embedding space, moving selected constant samples, into the neighborhood of sinusoidal/increasing/decreasing samples.

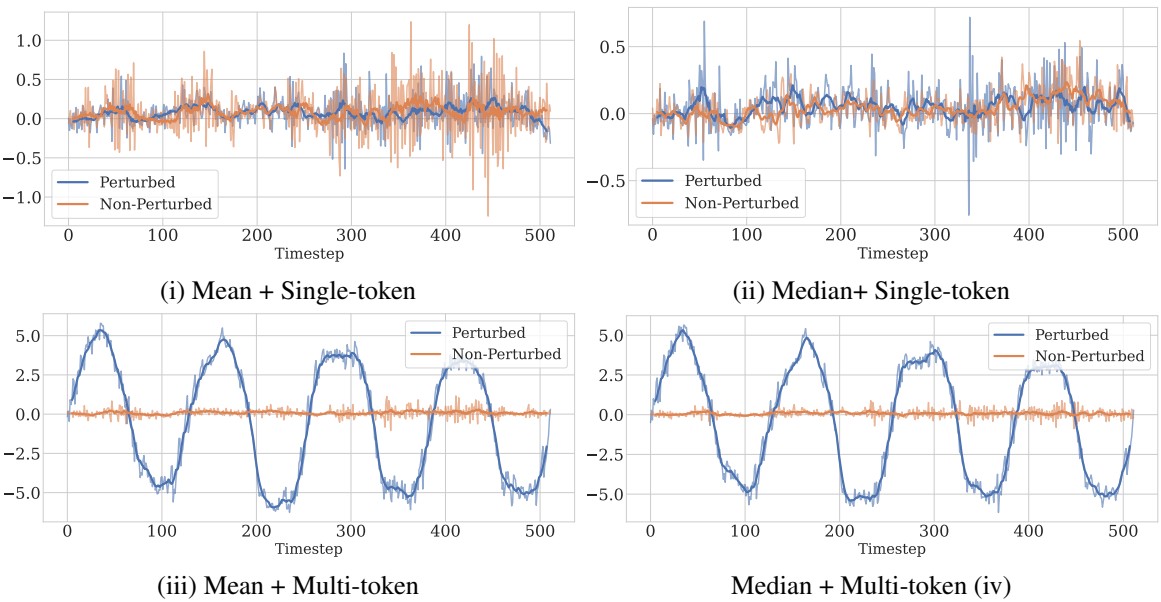

(i) Mean + Single-token      (ii) Median+ Single-token

(iii) Mean + Multi-token      Median + Multi-token (iv)

Figure 12: Visualization of different intervention and steering matrix derivation techniques. Applying concept steering interventions across all tokens is necessary to achieve the intended steered concept output (iii, iv) compared to applying concept steering interventions to a single token (i, iii). The method of obtaining the steering matrix, either by computing the mean or median across embedding concept classes, has no notable effect on the steered output.

**Application of Concept Steering on a Real-World Dataset**   We demonstrate the practical utility of concept steering on the ECG5000 dataset, which contains electrocardiogram readings classified as either normal or abnormal heart patterns. Using a `MOMENT` with an SVM classifier, we achieve strong baseline performance in distinguishing between the two classes.

For the steering experiment, we compute steering matrices using the median method to capture the concept difference between normal and abnormal patterns. Analysis in the activation space reveals that our steering approach successfully moves samples toward the target concept (Figure 13). This bidirectional movement is evident the PCA visualization of activation patterns.

To validate our approach quantitatively, we applied the steering transformations to 30 samples. While these samples were initially classified correctly with 100% accuracy as normal heartbeats, after steering, all of these samples swapped output classes, confirming successful concept transfer. These results suggest that concept steering can effectively capture and manipulate clinically relevant patterns in physiological data.

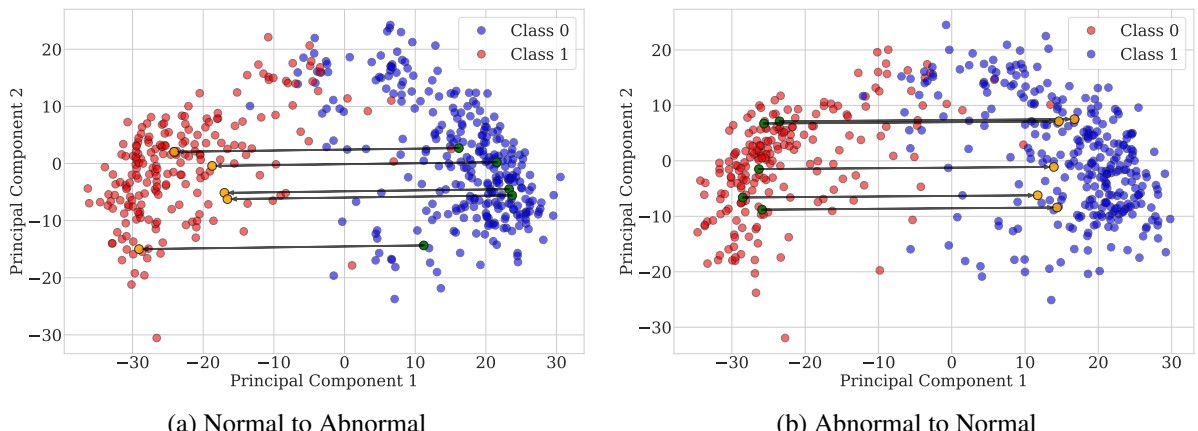

(a) Normal to Abnormal          (b) Abnormal to Normal

Figure 13: **Visualization of the impact of concept steering on the activation space of the ECG data model.** The figure shows PCA projections of the activation space, where arrows indicate the direction of steering. Blue and red points represent normal (class 0) and abnormal (class 1) samples respectively, with green points showing the original samples and orange points showing their steered versions.

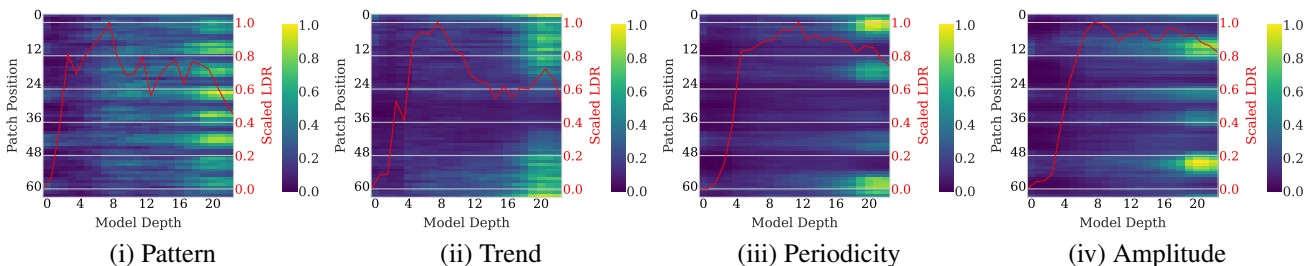

(i) Pattern        (ii) Trend        (iii) Periodicity        (iv) Amplitude

Figure 14: Linear separability of concepts at the patch level (y-axis) across model layers (x-axis). Yellow indicates higher separability than blue in the colorbar. Linear separability is measured using the Linear Discriminant Ratio (LDR), derived from model embedding statistics for each predicted class: (i) constant vs. sinusoidal patterns, (ii) constant vs. increasing trend patterns, (iii) sinusoidal vs. increasing trend patterns and (iv) sinusoidal vs. decreasing trend patterns. Color gradient indicates separability, with dark blue representing low LDR and yellow indicating high LDR. *Certain concepts captured by MOMENT-Large exhibit linear separability, but this separability is not uniform across layers; instead, it emerges at specific points in the model.*

# G. Additional Pruning Results

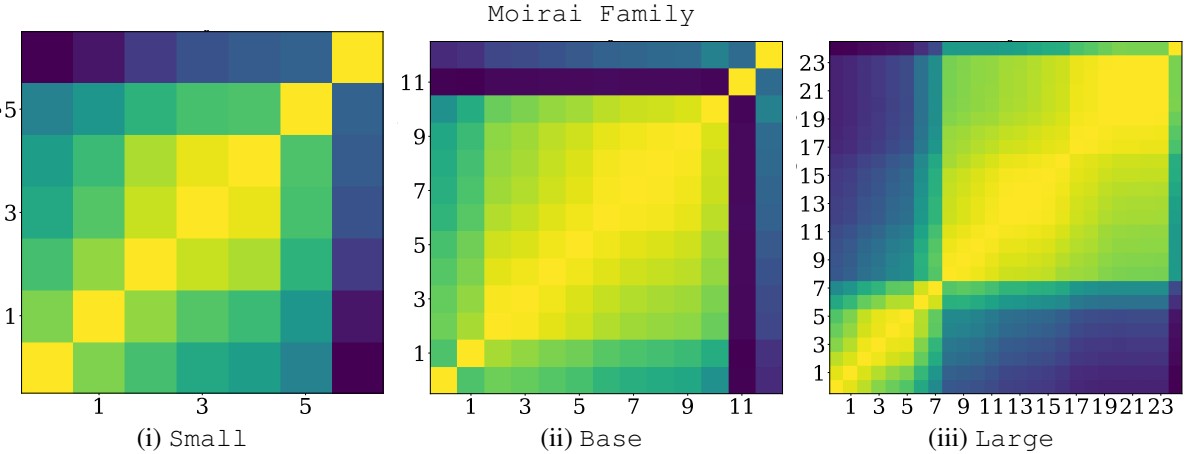

Figure 15: Self-similarity heatmaps comparing layer-to-layer CKA similarity matrices across different-sized models in the `Moirai` family. The Small and Base models exhibit similar patterns, with a distinct dissimilar layer at the end. The Base model has one additional dissimilar layer just before the final one. In contrast, the Large model presents a notably different structure, showing a two-block pattern where the layers are divided into two distinct groups, with the first block comprising roughly one-third of the model, and the second block covering the remaining two-thirds. This indicates a more pronounced hierarchical representation in the larger model.

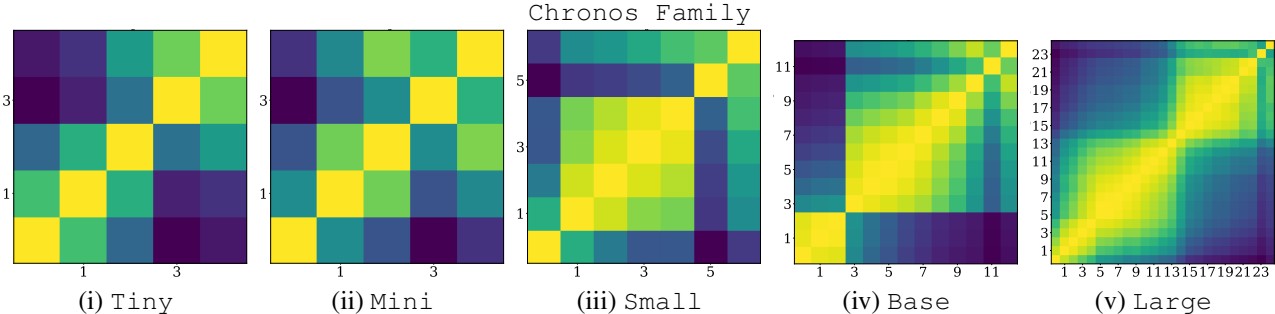

Figure 16: How does model size influence the patterns of learned representations? Smaller models exhibit more discrete and distinct block structures, while larger models (Base and Large) display increasingly intricate and clustered patterns, reflecting more nuanced and gradual transformations across layers. Notably, the emergence of blocks-like patterns in the `Large` model appears unpredictable from patterns observed in smaller models.

| Model name | Vanilla | | Block 1 | | Block 2 | | Block 3 | | All | |
|---|---|---|---|---|---|---|---|---|---|---|
| | MAE | MSE | MAE | MSE | MAE | MSE | MAE | MSE | MAE | MSE |
| ETTh1 | **0.395** | **0.371** | 0.448 | 0.449 | 0.521 | 0.614 | 0.420 | 0.424 | 0.548 | 0.673 |
| ETTh2 | **0.243** | **0.132** | 0.268 | 0.153 | 0.290 | 0.176 | **0.243** | 0.133 | 0.296 | 0.185 |
| ETTm1 | **0.287** | **0.204** | 0.321 | 0.233 | 0.358 | 0.307 | **0.287** | 0.221 | 0.345 | 0.284 |
| ETTm2 | 0.185 | 0.080 | 0.198 | 0.088 | 0.218 | 0.107 | **0.178** | **0.076** | 0.220 | 0.112 |
| Electricity | **0.372** | **0.250** | 0.446 | 0.342 | 0.716 | 0.757 | 0.428 | 0.327 | 0.727 | 0.819 |
| Exchange rate | 0.125 | 0.034 | 0.124 | 0.032 | 0.170 | 0.061 | **0.111** | **0.027** | 0.174 | 0.067 |
| Illness | **0.393** | **0.421** | 0.448 | 0.502 | 0.547 | 0.669 | 0.423 | 0.446 | 0.552 | 0.648 |
| Traffic | **0.492** | **0.790** | 0.552 | 0.906 | 0.861 | 1.562 | 0.606 | 0.940 | 0.878 | 1.633 |
| Weather | 0.129 | 0.079 | 0.134 | 0.082 | 0.170 | 0.107 | **0.117** | **0.074** | 0.176 | 0.120 |

Table 3: **Zero-shot imputation performance of MOMENT.** Results averaged across four different masking rates: {12.5%, 25%, 37.5%, 50%} and five runs with different masking seeds. This table presents the Model Performance Metrics (Mean Absolute Error and Mean Squared Error) on a subset of the Time Series Pile (Goswami et al., 2024; Zhou et al., 2021). The model names include: "Vanilla" MOMENT-Large without any pruning, "Block 1-3" for cases where only one block is pruned, and "All" for all three blocks being pruned. The best results per dataset are bolded. For full results refer to 5

| | Dataset | Vanilla | | Block 1 | | Block 2 | | Block 3 | | Block 4 | | All | |
|---|---|---|---|---|---|---|---|---|---|---|---|---|---|
| | | MASE | WQL | MASE | WQL | MASE | WQL | MASE | WQL | MASE | WQL | MASE | WQL |
| 0 | ETTh | 0.776 | **0.076** | 0.848 | 0.088 | **0.765** | 0.078 | 0.778 | 0.079 | 0.899 | 0.098 | 1.093 | 0.100 |
| 1 | ETTm | **0.716** | 0.068 | 0.820 | 0.076 | 0.864 | 0.084 | 0.721 | **0.063** | 0.724 | 0.074 | 0.978 | 0.084 |
| 2 | dominick | 0.820 | **0.331** | 0.829 | 0.335 | 0.824 | 0.334 | **0.809** | 0.332 | 0.835 | 0.337 | 0.886 | 0.359 |
| 3 | ercot | 0.627 | 0.020 | 0.694 | 0.022 | 0.697 | 0.027 | **0.624** | **0.018** | 0.909 | 0.024 | 1.281 | 0.042 |
| 4 | exchange_rate | 2.310 | **0.012** | 2.394 | 0.014 | 2.061 | 0.013 | 2.256 | 0.014 | 1.877 | 0.012 | **1.713** | 0.013 |
| 5 | m4_quarterly | 1.217 | 0.082 | 1.255 | 0.084 | **1.193** | **0.081** | 1.223 | 0.082 | 1.213 | 0.082 | 1.270 | 0.085 |
| 6 | m4_yearly | 3.559 | 0.133 | 3.537 | 0.132 | 3.495 | 0.129 | 3.507 | 0.131 | 3.502 | 0.129 | **3.359** | **0.123** |
| 7 | m5 | 0.943 | **0.586** | 0.942 | 0.586 | 0.954 | 0.591 | **0.939** | 0.587 | 0.941 | 0.622 | 0.956 | 0.644 |
| 8 | monash_australian_electricity | 1.427 | 0.076 | 1.417 | 0.077 | **1.372** | **0.073** | 1.445 | 0.083 | 1.638 | 0.084 | 2.593 | 0.143 |
| 9 | monash_car_parts | 0.903 | 1.041 | 0.883 | 1.046 | 0.867 | 0.998 | 0.892 | 1.027 | 0.833 | 0.960 | **0.816** | **0.956** |
| 10 | monash_cif_2016 | 0.989 | 0.012 | 1.058 | 0.019 | **0.944** | 0.013 | 0.998 | 0.018 | 1.096 | **0.011** | 1.275 | 0.017 |
| 11 | monash_covid_deaths | 43.251 | 0.058 | 42.444 | **0.052** | 44.001 | 0.078 | 42.357 | 0.060 | **41.915** | 0.053 | 45.544 | 0.062 |
| 12 | monash_fred_md | 0.517 | 0.021 | 0.520 | 0.019 | 0.507 | 0.024 | 0.553 | 0.029 | **0.504** | **0.016** | 0.600 | 0.017 |
| 13 | monash_hospital | 0.704 | 0.056 | 0.724 | 0.057 | **0.690** | **0.055** | 0.703 | 0.055 | 0.718 | 0.058 | 0.726 | 0.060 |
| 14 | monash_m1_monthly | 1.075 | 0.127 | 1.117 | 0.132 | **1.057** | **0.125** | 1.107 | 0.130 | 1.222 | 0.141 | 1.321 | 0.155 |
| 15 | monash_m1_quarterly | 1.728 | 0.106 | 1.743 | 0.102 | **1.659** | 0.105 | 1.727 | **0.092** | 1.748 | 0.105 | 1.753 | 0.098 |
| 16 | monash_m1_yearly | 4.336 | 0.183 | 4.275 | 0.174 | 4.113 | 0.174 | 4.375 | **0.163** | 4.400 | 0.182 | **4.100** | 0.170 |
| 17 | monash_m3_monthly | 0.855 | 0.096 | 0.887 | 0.100 | **0.845** | **0.094** | 0.864 | 0.096 | 0.909 | 0.102 | 0.968 | 0.108 |
| 18 | monash_m3_quarterly | 1.183 | 0.075 | 1.204 | 0.075 | **1.178** | **0.073** | 1.199 | 0.075 | 1.204 | 0.074 | 1.301 | 0.077 |
| 19 | monash_m3_yearly | 3.034 | 0.145 | 2.972 | 0.146 | 2.908 | 0.143 | 2.980 | 0.143 | 3.017 | 0.144 | **2.870** | **0.137** |
| 20 | monash_nn5_weekly | 0.936 | **0.090** | 0.961 | 0.092 | **0.924** | 0.090 | 0.939 | 0.090 | 0.941 | 0.090 | 0.985 | 0.095 |
| 21 | monash_tourism_monthly | 1.746 | 0.099 | 1.847 | 0.101 | **1.613** | **0.088** | 1.996 | 0.103 | 2.178 | 0.191 | 2.970 | 0.217 |
| 22 | monash_tourism_quarterly | 1.647 | 0.072 | 1.729 | 0.065 | **1.615** | **0.059** | 1.666 | 0.063 | 2.011 | 0.072 | 2.034 | 0.071 |
| 23 | monash_tourism_yearly | **3.565** | 0.180 | 3.683 | 0.185 | 3.574 | **0.173** | 3.724 | 0.188 | 3.596 | 0.174 | 3.568 | 0.175 |
| 24 | monash_traffic | **0.794** | 0.253 | 0.808 | **0.247** | 0.797 | 0.253 | 0.808 | 0.255 | 0.916 | 0.275 | 1.046 | 0.290 |
| 25 | monash_weather | **0.819** | **0.139** | 0.900 | 0.153 | 0.826 | 0.140 | 0.845 | 0.144 | 1.013 | 0.173 | 1.002 | 0.170 |
| 26 | nn5 | 0.574 | 0.157 | 0.584 | 0.160 | 0.599 | 0.165 | **0.571** | **0.154** | 0.792 | 0.219 | 0.905 | 0.255 |

Table 4: **Zero-shot forecasting performance of Chronos-Large.** This table presents zero-shot performance evaluated with Mean Absolute Scaled Error (MASE) and Weighted Quantile Loss (WQL). Results are presented for the model without any pruning (Vanilla), when individual blocks pruned Block i, and when all blocks are pruned (All).

| Dataset | Mask Ratio | All Mean MAE | All Mean MSE | All Std. MAE | All Std. MSE | Block 1 Mean MAE | Block 1 Mean MSE | Block 1 Std. MAE | Block 1 Std. MSE | Block 2 Mean MAE | Block 2 Mean MSE | Block 2 Std. MAE | Block 2 Std. MSE | Block 3 Mean MAE | Block 3 Mean MSE | Block 3 Std. MAE | Block 3 Std. MSE | Vanilla Mean MAE | Vanilla Mean MSE | Vanilla Std. MAE | Vanilla Std. MSE |
|---|---|---|---|---|---|---|---|---|---|---|---|---|---|---|---|---|---|---|---|---|---|
| ETTh1 | 0.125 | 0.541 | 0.662 | 0.016 | 0.069 | 0.448 | 0.446 | 0.023 | 0.084 | 0.516 | 0.608 | 0.016 | 0.072 | 0.426 | 0.425 | 0.038 | 0.119 | 0.398 | 0.370 | 0.034 | 0.091 |
|  | 0.25 | 0.548 | 0.667 | 0.015 | 0.049 | 0.445 | 0.441 | 0.019 | 0.056 | 0.520 | 0.605 | 0.021 | 0.073 | 0.418 | 0.420 | 0.026 | 0.073 | 0.393 | 0.365 | 0.026 | 0.061 |
|  | 0.375 | 0.550 | 0.674 | 0.015 | 0.035 | 0.447 | 0.446 | 0.011 | 0.033 | 0.522 | 0.612 | 0.016 | 0.045 | 0.418 | 0.422 | 0.019 | 0.056 | 0.394 | 0.368 | 0.014 | 0.033 |
|  | 0.5 | 0.551 | 0.687 | 0.011 | 0.036 | 0.452 | 0.463 | 0.012 | 0.037 | 0.525 | 0.630 | 0.011 | 0.041 | 0.420 | 0.429 | 0.011 | 0.035 | 0.397 | 0.380 | 0.010 | 0.027 |
|  | mean | 0.548 | 0.673 | 0.014 | 0.046 | 0.448 | 0.449 | 0.016 | 0.052 | 0.521 | 0.614 | 0.015 | 0.056 | 0.420 | 0.424 | 0.024 | 0.071 | 0.395 | 0.371 | 0.021 | 0.054 |
| ETTh2 | 0.125 | 0.290 | 0.179 | 0.005 | 0.016 | 0.269 | 0.155 | 0.008 | 0.016 | 0.292 | 0.177 | 0.010 | 0.020 | 0.240 | 0.130 | 0.010 | 0.015 | 0.243 | 0.131 | 0.007 | 0.015 |
|  | 0.25 | 0.298 | 0.188 | 0.008 | 0.015 | 0.271 | 0.159 | 0.011 | 0.019 | 0.294 | 0.182 | 0.010 | 0.019 | 0.245 | 0.136 | 0.010 | 0.015 | 0.247 | 0.136 | 0.012 | 0.017 |
|  | 0.375 | 0.296 | 0.186 | 0.005 | 0.009 | 0.266 | 0.151 | 0.011 | 0.015 | 0.286 | 0.172 | 0.008 | 0.013 | 0.245 | 0.135 | 0.007 | 0.010 | 0.242 | 0.130 | 0.011 | 0.014 |
|  | 0.5 | 0.298 | 0.187 | 0.005 | 0.010 | 0.266 | 0.150 | 0.008 | 0.011 | 0.288 | 0.174 | 0.006 | 0.011 | 0.242 | 0.132 | 0.006 | 0.008 | 0.242 | 0.129 | 0.007 | 0.010 |
|  | mean | 0.296 | 0.185 | 0.006 | 0.012 | 0.268 | 0.153 | 0.009 | 0.015 | 0.290 | 0.176 | 0.008 | 0.015 | 0.243 | 0.133 | 0.008 | 0.012 | 0.243 | 0.132 | 0.009 | 0.013 |
| ETTm1 | 0.125 | 0.344 | 0.281 | 0.013 | 0.039 | 0.322 | 0.236 | 0.012 | 0.023 | 0.361 | 0.313 | 0.016 | 0.035 | 0.284 | 0.216 | 0.008 | 0.026 | 0.288 | 0.208 | 0.011 | 0.025 |
|  | 0.25 | 0.343 | 0.280 | 0.010 | 0.026 | 0.319 | 0.230 | 0.012 | 0.026 | 0.357 | 0.304 | 0.012 | 0.029 | 0.284 | 0.216 | 0.008 | 0.024 | 0.284 | 0.199 | 0.013 | 0.028 |
|  | 0.375 | 0.345 | 0.283 | 0.005 | 0.014 | 0.319 | 0.230 | 0.005 | 0.013 | 0.356 | 0.302 | 0.006 | 0.016 | 0.288 | 0.222 | 0.004 | 0.011 | 0.286 | 0.199 | 0.006 | 0.016 |
|  | 0.5 | 0.347 | 0.292 | 0.004 | 0.010 | 0.322 | 0.238 | 0.004 | 0.009 | 0.357 | 0.309 | 0.002 | 0.006 | 0.291 | 0.229 | 0.002 | 0.010 | 0.288 | 0.207 | 0.003 | 0.008 |
|  | mean | 0.345 | 0.284 | 0.009 | 0.024 | 0.321 | 0.233 | 0.009 | 0.018 | 0.358 | 0.307 | 0.010 | 0.023 | 0.287 | 0.221 | 0.006 | 0.018 | 0.287 | 0.204 | 0.008 | 0.020 |
| ETTm2 | 0.125 | 0.222 | 0.113 | 0.005 | 0.008 | 0.200 | 0.089 | 0.004 | 0.002 | 0.221 | 0.110 | 0.005 | 0.004 | 0.179 | 0.078 | 0.005 | 0.004 | 0.188 | 0.082 | 0.005 | 0.004 |
|  | 0.25 | 0.220 | 0.112 | 0.006 | 0.005 | 0.197 | 0.086 | 0.005 | 0.004 | 0.216 | 0.105 | 0.006 | 0.005 | 0.177 | 0.074 | 0.004 | 0.003 | 0.184 | 0.078 | 0.004 | 0.003 |
|  | 0.375 | 0.219 | 0.112 | 0.003 | 0.003 | 0.198 | 0.088 | 0.003 | 0.002 | 0.216 | 0.106 | 0.004 | 0.003 | 0.178 | 0.075 | 0.002 | 0.002 | 0.184 | 0.080 | 0.003 | 0.002 |
|  | 0.5 | 0.219 | 0.111 | 0.001 | 0.002 | 0.198 | 0.088 | 0.003 | 0.003 | 0.218 | 0.107 | 0.004 | 0.004 | 0.178 | 0.075 | 0.002 | 0.001 | 0.185 | 0.080 | 0.003 | 0.003 |
|  | mean | 0.220 | 0.112 | 0.004 | 0.005 | 0.198 | 0.088 | 0.004 | 0.003 | 0.218 | 0.107 | 0.005 | 0.004 | 0.178 | 0.076 | 0.003 | 0.003 | 0.185 | 0.080 | 0.004 | 0.003 |
| electricity | 0.125 | 0.736 | 0.838 | 0.019 | 0.037 | 0.449 | 0.345 | 0.009 | 0.011 | 0.722 | 0.769 | 0.013 | 0.025 | 0.425 | 0.321 | 0.010 | 0.015 | 0.375 | 0.253 | 0.012 | 0.014 |
|  | 0.25 | 0.725 | 0.816 | 0.011 | 0.021 | 0.443 | 0.338 | 0.006 | 0.007 | 0.715 | 0.755 | 0.009 | 0.016 | 0.428 | 0.325 | 0.004 | 0.007 | 0.371 | 0.249 | 0.006 | 0.007 |
|  | 0.375 | 0.722 | 0.809 | 0.007 | 0.012 | 0.444 | 0.341 | 0.002 | 0.003 | 0.712 | 0.750 | 0.005 | 0.008 | 0.430 | 0.331 | 0.003 | 0.003 | 0.369 | 0.248 | 0.004 | 0.004 |
|  | 0.5 | 0.725 | 0.814 | 0.002 | 0.002 | 0.447 | 0.345 | 0.004 | 0.004 | 0.714 | 0.754 | 0.002 | 0.004 | 0.431 | 0.332 | 0.003 | 0.003 | 0.371 | 0.249 | 0.002 | 0.002 |
|  | mean | 0.727 | 0.819 | 0.012 | 0.023 | 0.446 | 0.342 | 0.006 | 0.007 | 0.716 | 0.757 | 0.009 | 0.016 | 0.428 | 0.327 | 0.006 | 0.009 | 0.372 | 0.250 | 0.007 | 0.008 |
| exchange rate | 0.125 | 0.164 | 0.059 | 0.010 | 0.014 | 0.129 | 0.035 | 0.007 | 0.006 | 0.178 | 0.065 | 0.010 | 0.008 | 0.109 | 0.027 | 0.010 | 0.002 | 0.131 | 0.037 | 0.010 | 0.006 |
|  | 0.25 | 0.172 | 0.064 | 0.010 | 0.012 | 0.122 | 0.031 | 0.006 | 0.003 | 0.167 | 0.059 | 0.001 | 0.002 | 0.108 | 0.025 | 0.007 | 0.001 | 0.122 | 0.033 | 0.008 | 0.004 |
|  | 0.375 | 0.178 | 0.070 | 0.005 | 0.009 | 0.122 | 0.032 | 0.004 | 0.002 | 0.167 | 0.059 | 0.008 | 0.004 | 0.112 | 0.028 | 0.009 | 0.004 | 0.123 | 0.033 | 0.008 | 0.004 |
|  | 0.5 | 0.180 | 0.073 | 0.008 | 0.014 | 0.123 | 0.032 | 0.007 | 0.003 | 0.170 | 0.061 | 0.009 | 0.006 | 0.114 | 0.029 | 0.005 | 0.002 | 0.125 | 0.034 | 0.011 | 0.006 |
|  | mean | 0.174 | 0.067 | 0.010 | 0.013 | 0.124 | 0.032 | 0.006 | 0.004 | 0.170 | 0.061 | 0.008 | 0.006 | 0.111 | 0.027 | 0.008 | 0.003 | 0.125 | 0.034 | 0.009 | 0.005 |
| national illness | 0.125 | 0.612 | 0.840 | 0.122 | 0.477 | 0.521 | 0.722 | 0.157 | 0.597 | 0.619 | 0.916 | 0.164 | 0.660 | 0.454 | 0.590 | 0.129 | 0.522 | 0.443 | 0.588 | 0.145 | 0.541 |
|  | 0.25 | 0.563 | 0.662 | 0.066 | 0.205 | 0.457 | 0.509 | 0.078 | 0.283 | 0.552 | 0.674 | 0.069 | 0.284 | 0.425 | 0.449 | 0.060 | 0.246 | 0.403 | 0.430 | 0.080 | 0.255 |
|  | 0.375 | 0.512 | 0.545 | 0.039 | 0.148 | 0.408 | 0.403 | 0.056 | 0.204 | 0.501 | 0.547 | 0.054 | 0.208 | 0.407 | 0.383 | 0.036 | 0.177 | 0.364 | 0.348 | 0.057 | 0.183 |
|  | 0.5 | 0.521 | 0.544 | 0.019 | 0.084 | 0.406 | 0.374 | 0.031 | 0.139 | 0.516 | 0.538 | 0.032 | 0.147 | 0.407 | 0.361 | 0.028 | 0.119 | 0.360 | 0.320 | 0.040 | 0.130 |
|  | mean | 0.552 | 0.648 | 0.078 | 0.279 | 0.448 | 0.502 | 0.098 | 0.353 | 0.547 | 0.669 | 0.098 | 0.383 | 0.423 | 0.446 | 0.071 | 0.297 | 0.393 | 0.421 | 0.089 | 0.312 |
| traffic | 0.125 | 0.880 | 1.647 | 0.018 | 0.052 | 0.559 | 0.915 | 0.023 | 0.060 | 0.860 | 1.572 | 0.015 | 0.028 | 0.609 | 0.950 | 0.027 | 0.068 | 0.503 | 0.803 | 0.025 | 0.068 |
|  | 0.25 | 0.887 | 1.660 | 0.015 | 0.032 | 0.550 | 0.909 | 0.008 | 0.054 | 0.871 | 1.589 | 0.016 | 0.018 | 0.602 | 0.936 | 0.023 | 0.065 | 0.492 | 0.793 | 0.014 | 0.056 |
|  | 0.375 | 0.874 | 1.620 | 0.011 | 0.034 | 0.548 | 0.901 | 0.012 | 0.052 | 0.860 | 1.553 | 0.011 | 0.028 | 0.603 | 0.931 | 0.015 | 0.043 | 0.487 | 0.780 | 0.006 | 0.044 |
|  | 0.5 | 0.869 | 1.603 | 0.012 | 0.037 | 0.549 | 0.898 | 0.013 | 0.046 | 0.854 | 1.534 | 0.012 | 0.035 | 0.608 | 0.943 | 0.007 | 0.021 | 0.487 | 0.783 | 0.006 | 0.035 |
|  | mean | 0.878 | 1.633 | 0.015 | 0.043 | 0.552 | 0.906 | 0.015 | 0.049 | 0.861 | 1.562 | 0.014 | 0.033 | 0.606 | 0.940 | 0.018 | 0.049 | 0.492 | 0.790 | 0.015 | 0.049 |
| weather | 0.125 | 0.175 | 0.121 | 0.004 | 0.007 | 0.134 | 0.082 | 0.003 | 0.006 | 0.171 | 0.108 | 0.004 | 0.005 | 0.115 | 0.071 | 0.005 | 0.009 | 0.130 | 0.078 | 0.004 | 0.007 |
|  | 0.25 | 0.176 | 0.120 | 0.002 | 0.005 | 0.134 | 0.081 | 0.005 | 0.004 | 0.169 | 0.105 | 0.006 | 0.007 | 0.116 | 0.073 | 0.002 | 0.003 | 0.128 | 0.077 | 0.004 | 0.004 |
|  | 0.375 | 0.177 | 0.121 | 0.001 | 0.002 | 0.134 | 0.083 | 0.005 | 0.007 | 0.168 | 0.107 | 0.005 | 0.006 | 0.117 | 0.076 | 0.003 | 0.006 | 0.129 | 0.080 | 0.005 | 0.007 |
|  | 0.5 | 0.176 | 0.119 | 0.002 | 0.003 | 0.135 | 0.084 | 0.004 | 0.007 | 0.170 | 0.107 | 0.004 | 0.006 | 0.118 | 0.076 | 0.003 | 0.006 | 0.130 | 0.081 | 0.004 | 0.007 |
|  | mean | 0.176 | 0.120 | 0.002 | 0.004 | 0.134 | 0.082 | 0.004 | 0.006 | 0.170 | 0.107 | 0.005 | 0.006 | 0.117 | 0.074 | 0.003 | 0.006 | 0.129 | 0.079 | 0.004 | 0.006 |

Table 5: **Zero-shot imputation performance of MOMENT.** Results averaged across five runs with different masking seeds. This table presents the Model Performance Metrics (Mean Absolute Error and Mean Squared Error) on a subset of the Time Series Pile (Goswami et al., 2024; Zhou et al., 2021). The model names include: "Vanilla" MOMENT-Large without any pruning, "Block 1-3" for cases where only one block is pruned, and "All" for all three blocks being pruned.

| Dataset | Horizon | | | | Horizon | | | |
|---|---|---|---|---|---|---|---|---|
| | 96 | 192 | 336 | 720 | 96 | 192 | 336 | 720 |
| | **Vanilla MAE** | | | | **All Pruned MAE** | | | |
| ETTh1 | **0.409** | **0.426** | **0.437** | **0.464** | **0.409** | **0.426** | **0.437** | 0.470 |
| ETTh2 | **0.346** | **0.386** | **0.409** | 0.440 | 0.351 | 0.389 | 0.413 | **0.439** |
| ETTm1 | 0.347 | 0.373 | **0.386** | **0.420** | **0.346** | **0.369** | **0.386** | 0.422 |
| ETTm2 | **0.260** | **0.300** | **0.337** | **0.392** | 0.262 | 0.304 | 0.343 | 0.395 |
| Weather | **0.209** | **0.248** | **0.285** | **0.337** | **0.209** | **0.248** | 0.287 | **0.337** |
| | **Vanilla MSE** | | | | **All Pruned MSE** | | | |
| ETTh1 | **0.385** | **0.411** | **0.423** | **0.443** | 0.388 | 0.414 | 0.424 | 0.460 |
| ETTh2 | **0.287** | **0.350** | **0.370** | **0.404** | 0.296 | 0.356 | 0.382 | **0.404** |
| ETTm1 | **0.290** | 0.330 | **0.352** | **0.409** | 0.29 | 0.326 | 0.354 | 0.414 |
| ETTm2 | **0.171** | **0.231** | **0.287** | **0.372** | 0.173 | 0.236 | 0.294 | **0.372** |
| Weather | 0.153 | **0.197** | **0.246** | **0.316** | **0.152** | 0.198 | 0.247 | 0.317 |

Table 6: **Fine-tuned forecasting performance of MOMENT.** This table presents model performance metrics (Mean Absolute Error and Mean Squared Error) on a subset of the Time Series Pile (Goswami et al., 2024; Zhou et al., 2021). Metrics are presented for MOMENT-Large without any pruning (Vanilla) and for all blocks pruned (All). Results are gathered from the best performance on the test set across 3 epochs of training with a batch size 64 and learning rate of 0.0001.

| Method | Pruned Part | Encoder Sparsity (%) | Model Sparsity (%) | Estimated Model Size (MB) | Average Time (ms) | Standard Deviation (ms) |
|---|---|---|---|---|---|---|
| MOMENT | None | 0.00 | 0.00 | 1301.76 | 20.88 | 0.42 |
| | Block 1 | 12.50 | 11.29 | 1154.76 | 19.42 | 0.27 |
| | Block 2 | 33.33 | 30.11 | 909.76 | 19.71 | 0.31 |
| | Block 3 | 12.50 | 11.29 | 1154.76 | 19.56 | 0.29 |
| | All Blocks | 58.33 | 52.70 | 615.76 | 19.82 | 0.43 |
| Chronos-T5 | None | 0.00 | 0.00 | 2704.48 | 59.95 | 0.26 |
| | Block 1 | 8.33 | 3.55 | 2608.48 | 58.22 | 0.58 |
| | Block 2 | 12.50 | 5.32 | 2560.48 | 56.80 | 0.78 |
| | Block 3 | 8.33 | 3.55 | 2608.48 | 56.94 | 0.27 |
| | Block 4 | 25.00 | 10.65 | 2416.48 | 57.55 | 0.84 |
| | All Blocks | 54.17 | 23.07 | 2080.48 | 56.82 | 0.58 |

Table 7: **Inference performance under various pruning configurations.** This table presents the inference performance metrics of the MOMENT-Large and Chronos-Large models with different pruning configurations. Results are presented for MOMENT-Large and Chronos without any pruning (None), when individual blocks pruned Block i, and when all blocks are pruned (All Blocks). Inference time estimation was performed by aggregating times from 100 passes of a one-batch, one-channel sample of length 512.

| Statistic Category | Metric | Mean Difference | Standard Deviation | Original Better | Pruned Better | Equal | Pearson Correlation Coefficient |
|---|---|---|---|---|---|---|---|
| **Classification** | Accuracy | 0.0128 | 0.0489 | 54 | 28 | 9 | 0.945 |
| **Anomaly Detection** | Adjusted Best $F_1$ | 0.2117 | 0.3026 | 28 | 7 | 6 | 0.634 |
| | VUS-ROC | 0.1007 | 0.0995 | 36 | 2 | 3 | 0.512 |

Table 8: **Summary of differences between MOMENT_pruned (all blocks pruned) and MOMENT.** Detailed results are provided in Tables 9 and 11. For classification, we observed a slight deterioration in performance, whereas for anomaly detection the deterioration was more severe.

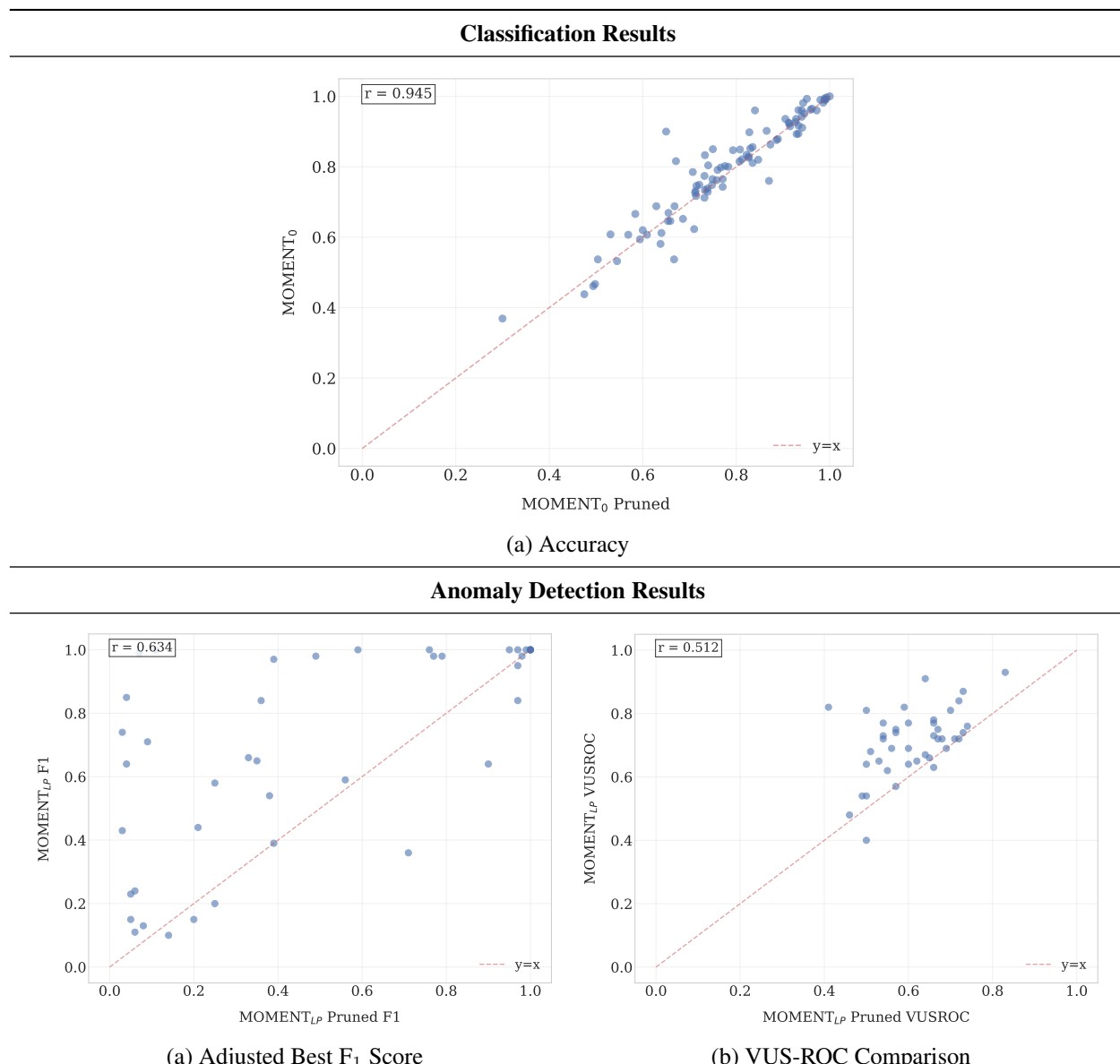

Figure 17: **Combined visualization of classification accuracy and anomaly detection metrics** (F1 Score and VUS-ROC) comparing pruned and non-pruned models. Classification results are based on 91 UCR datasets ($MOMENT_0$ and $MOMENT_{0\_pruned}$), where the mean accuracy is 79.4% and 78.1% respectively (see Table 10). Anomaly detection uses a subset of 44 datasets from the UCR Anomaly Archive ($MOMENT_{LP}$ and $MOMENT_{LP\_pruned}$). See Tables 11 and 9 for detailed results.

| Model name / Dataset name | Adj. Best F1 | | | | | | | VUSROC | | | | | | |
|---|---|---|---|---|---|---|---|---|---|---|---|---|---|---|
| | AnomalyNearestNeighbors | AnomalyTransformer | MOMENT$_{LP}$ | MOMENT$_{LP\_pruned}$ | DGHL | GPT4TS | TimesNet | AnomalyNearestNeighbors | AnomalyTransformer | MOMENT$_{LP}$ | MOMENT$_{LP\_pruned}$ | DGHL | GPT4TS | TimesNet |
| 1sddb40 | 0.720 | 0.030 | 0.540 | 0.380 | 0.390 | 0.190 | 0.680 | 0.680 | 0.640 | 0.750 | 0.570 | 0.640 | 0.660 | 0.720 |
| BIDMC1 | 1.000 | 0.990 | 1.000 | 0.950 | 1.000 | 1.000 | 1.000 | 0.660 | 0.690 | 0.650 | 0.620 | 0.720 | 0.630 | 0.740 |
| CHARISfive | 0.090 | 0.010 | 0.130 | 0.080 | 0.020 | 0.020 | 0.080 | 0.830 | 0.360 | 0.400 | 0.500 | 0.510 | 0.450 | 0.460 |
| CHARISten | 0.020 | 0.020 | 0.110 | 0.060 | 0.040 | 0.100 | 0.030 | 0.520 | 0.430 | 0.540 | 0.500 | 0.520 | 0.510 | 0.530 |
| CIMIS44AirTemperature3 | 1.000 | 0.060 | 0.980 | 0.490 | 0.500 | 0.180 | 0.470 | 0.860 | 0.640 | 0.750 | 0.670 | 0.740 | 0.620 | 0.740 |
| CIMIS44AirTemperature5 | 0.990 | 0.390 | 0.990 | 0.070 | 0.960 | 0.200 | 0.710 | 0.900 | 0.780 | 0.810 | 0.500 | 0.920 | 0.560 | 0.720 |
| ECG2 | 0.860 | 1.000 | 1.000 | 1.000 | 0.620 | 0.900 | 1.000 | 0.840 | 0.830 | 0.840 | 0.720 | 0.630 | 0.780 | 0.600 |
| ECG3 | 1.000 | 0.360 | 0.980 | 0.770 | 0.800 | 0.840 | 0.480 | 0.760 | 0.540 | 0.770 | 0.540 | 0.680 | 0.450 | 0.610 |
| Fantasia | 0.770 | 0.750 | 0.950 | 0.970 | 0.660 | 0.870 | 0.550 | 0.610 | 0.730 | 0.640 | 0.500 | 0.710 | 0.650 | 0.610 |
| GP711MarkerLFM5z4 | 1.000 | 0.930 | 1.000 | 0.760 | 0.500 | 0.640 | 0.950 | 0.790 | 0.540 | 0.730 | 0.540 | 0.600 | 0.620 | 0.720 |
| GP711MarkerLFM5z5 | 1.000 | 0.760 | 0.970 | 0.390 | 0.310 | 0.480 | 0.900 | 0.980 | 0.690 | 0.720 | 0.670 | 0.520 | 0.630 | 0.840 |
| InternalBleeding5 | 0.910 | 0.940 | 1.000 | 0.990 | 1.000 | 0.920 | 1.000 | 0.880 | 0.460 | 0.690 | 0.690 | 0.760 | 0.630 | 0.940 |
| Italianpowerdemand | 0.060 | 0.010 | 0.740 | 0.030 | 0.590 | 0.010 | 0.440 | 0.630 | 0.450 | 0.770 | 0.600 | 0.700 | 0.480 | 0.710 |
| Lab2Cmac011215EPG5 | 0.460 | 0.990 | 0.980 | 0.980 | 0.340 | 0.600 | 0.990 | 0.760 | 0.770 | 0.630 | 0.660 | 0.710 | 0.640 | 0.610 |
| Lab2Cmac011215EPG6 | 0.240 | 0.410 | 0.100 | 0.140 | 0.260 | 0.100 | 0.170 | 0.600 | 0.700 | 0.480 | 0.460 | 0.600 | 0.520 | 0.450 |
| MesoplodonDensirostris | 1.000 | 1.000 | 0.840 | 0.970 | 0.790 | 1.000 | 1.000 | 0.780 | 0.850 | 0.720 | 0.710 | 0.740 | 0.690 | 0.790 |
| PowerDemand1 | 0.800 | 0.870 | 0.440 | 0.210 | 0.490 | 0.760 | 0.950 | 0.800 | 0.720 | 0.540 | 0.490 | 0.530 | 0.600 | 0.750 |
| TkeepFirstMARS | 0.400 | 0.010 | 0.150 | 0.200 | 0.020 | 0.020 | 0.230 | 0.510 | 0.520 | 0.760 | 0.740 | 0.460 | 0.500 | 0.790 |
| TkeepSecondMARS | 0.950 | 0.830 | 1.000 | 1.000 | 0.160 | 0.120 | 0.950 | 0.750 | 0.720 | 0.910 | 0.640 | 0.970 | 0.810 | 0.980 |
| WalkingAceleration5 | 0.950 | 0.990 | 1.000 | 1.000 | 0.910 | 0.870 | 0.930 | 0.910 | 0.940 | 0.870 | 0.730 | 0.930 | 0.910 | 0.850 |
| apneaecg | 0.360 | 0.400 | 0.200 | 0.250 | 0.250 | 0.310 | 0.260 | 0.700 | 0.580 | 0.690 | 0.560 | 0.590 | 0.580 | 0.760 |
| apneaecg2 | 1.000 | 0.650 | 1.000 | 0.970 | 1.000 | 1.000 | 0.650 | 0.760 | 0.790 | 0.740 | 0.570 | 0.730 | 0.650 | 0.610 |
| gait1 | 1.000 | 0.180 | 0.360 | 0.710 | 0.070 | 0.410 | 0.520 | 0.640 | 0.630 | 0.570 | 0.570 | 0.600 | 0.580 | 0.600 |
| gaitHunt1 | 0.020 | 0.080 | 0.430 | 0.140 | 0.020 | 0.100 | 0.300 | 0.570 | 0.810 | 0.680 | 0.510 | 0.570 | 0.710 | 0.840 |
| insectEPG2 | 0.710 | 0.120 | 0.230 | 0.050 | 0.140 | 0.810 | 0.960 | 0.760 | 0.650 | 0.820 | 0.410 | 0.650 | 0.560 | 0.730 |
| insectEPG4 | 0.650 | 0.980 | 1.000 | 0.110 | 0.460 | 0.210 | 0.850 | 0.760 | 0.690 | 0.720 | 0.540 | 0.730 | 0.490 | 0.670 |
| ltstdbs30791AS | 1.000 | 1.000 | 1.000 | 1.000 | 1.000 | 1.000 | 1.000 | 0.760 | 0.780 | 0.810 | 0.700 | 0.770 | 0.740 | 0.670 |
| mit14046longtermecg | 0.600 | 0.450 | 0.590 | 0.560 | 0.530 | 0.580 | 0.600 | 0.720 | 0.790 | 0.660 | 0.650 | 0.640 | 0.610 | 0.840 |
| park3m | 0.550 | 0.150 | 0.640 | 0.900 | 0.200 | 0.630 | 0.930 | 0.730 | 0.630 | 0.780 | 0.660 | 0.650 | 0.540 | 0.780 |
| qtdbSel1005V | 0.390 | 0.410 | 0.650 | 0.350 | 0.400 | 0.390 | 0.530 | 0.550 | 0.520 | 0.640 | 0.600 | 0.490 | 0.610 | 0.540 |
| qtdbSel100MLII | 0.500 | 0.420 | 0.840 | 0.360 | 0.410 | 0.600 | 0.870 | 0.540 | 0.620 | 0.620 | 0.550 | 0.590 | 0.580 | 0.650 |
| resperation1 | 0.020 | 0.000 | 0.150 | 0.050 | 0.030 | 0.010 | 0.030 | 0.620 | 0.750 | 0.670 | 0.640 | 0.740 | 0.470 | 0.670 |
| s20101mML2 | 0.130 | 0.690 | 0.710 | 0.090 | 0.150 | 0.050 | 0.080 | 0.730 | 0.640 | 0.720 | 0.720 | 0.690 | 0.640 | 0.690 |
| sddb49 | 0.360 | 0.890 | 1.000 | 0.590 | 0.880 | 0.940 | 1.000 | 0.690 | 0.660 | 0.730 | 0.660 | 0.740 | 0.580 | 0.680 |
| sel840mECG1 | 0.350 | 0.160 | 0.660 | 0.330 | 0.280 | 0.210 | 0.360 | 0.740 | 0.620 | 0.720 | 0.680 | 0.870 | 0.650 | 0.600 |
| sel840mECG2 | 0.150 | 0.150 | 0.390 | 0.390 | 0.320 | 0.280 | 0.210 | 0.680 | 0.590 | 0.690 | 0.600 | 0.490 | 0.520 | 0.520 |
| tilt12744mtable | 0.060 | 0.070 | 0.240 | 0.060 | 0.100 | 0.000 | 0.030 | 0.690 | 0.480 | 0.740 | 0.730 | 0.660 | 0.510 | 0.640 |
| tilt12754table | 0.130 | 0.230 | 0.640 | 0.040 | 0.040 | 0.060 | 0.050 | 0.730 | 0.600 | 0.820 | 0.590 | 0.790 | 0.550 | 0.750 |
| tiltAPB2 | 0.690 | 0.920 | 0.980 | 0.790 | 0.360 | 0.830 | 0.380 | 0.710 | 0.770 | 0.770 | 0.660 | 0.710 | 0.600 | 0.700 |
| tiltAPB3 | 0.060 | 0.170 | 0.850 | 0.040 | 0.030 | 0.050 | 0.090 | 0.640 | 0.680 | 0.650 | 0.530 | 0.540 | 0.440 | 0.580 |
| weallwalk | 0.500 | 0.000 | 0.580 | 0.250 | 0.070 | 0.130 | 0.170 | 0.820 | 0.730 | 0.930 | 0.830 | 0.860 | 0.870 | 0.850 |

Table 9: **Anomaly detection** using Adjusted Best $F_1$ and VUS-ROC for a subset of 44 datasets sampled from the UCR Anomaly archive. MOMENT$_{LP}$ and pruned (all blocks) MOMENT$_{LP\_pruned}$.

| | MOMENT$_{0\_pruned}$ | MOMENT$_0$ |
|---|---|---|
| Mean | 0.781 | 0.794 |
| Std | 0.146 | 0.148 |
| Min | 0.300 | 0.369 |
| 25% | 0.697 | 0.714 |
| 50% | 0.776 | 0.815 |
| 75% | 0.915 | 0.916 |
| Max | 1.000 | 1.000 |

Table 10: **Statistic: Classification accuracy** of methods across 91 UCR datasets. MOMENT$_0$ without fine-tuning and pruned (all blocks) MOMENT$_{0\_pruned}$. See full table:11

| Dataset | MOMENT$_{0pruned}$ | MOMENT$_0$ | TS2Vec | T-Loss | TNC | TS-TCC | TST | DTW | CNN | Encoder | FCN | MCDNN | MLP | ResNet | t-LeNet | TWIESN |
|---|---|---|---|---|---|---|---|---|---|---|---|---|---|---|---|---|
| GestureMidAirD2 | 0.531 | 0.608 | 0.469 | 0.546 | 0.362 | 0.254 | 0.138 | 0.608 | 0.518 | 0.480 | 0.631 | 0.500 | 0.545 | 0.668 | 0.038 | 0.575 |
| UWaveGestureLibraryX | 0.812 | 0.821 | 0.795 | 0.785 | 0.781 | 0.733 | 0.569 | 0.728 | 0.721 | 0.771 | 0.754 | 0.726 | 0.768 | 0.781 | 0.127 | 0.608 |
| GesturePebbleZ2 | 0.671 | 0.816 | 0.873 | 0.899 | 0.316 | 0.430 | 0.380 | 0.671 | 0.778 | 0.796 | 0.781 | 0.720 | 0.701 | 0.777 | 0.184 | 0.843 |
| ECG5000 | 0.940 | 0.942 | 0.935 | 0.933 | 0.937 | 0.941 | 0.928 | 0.924 | 0.928 | 0.941 | 0.940 | 0.933 | 0.930 | 0.935 | 0.584 | 0.922 |
| OSULeaf | 0.707 | 0.785 | 0.851 | 0.760 | 0.723 | 0.723 | 0.545 | 0.591 | 0.482 | 0.554 | 0.979 | 0.419 | 0.560 | 0.980 | 0.182 | 0.628 |
| MedicalImages | 0.758 | 0.762 | 0.789 | 0.750 | 0.754 | 0.747 | 0.632 | 0.737 | 0.671 | 0.664 | 0.778 | 0.627 | 0.719 | 0.770 | 0.514 | 0.649 |
| Ham | 0.638 | 0.581 | 0.714 | 0.724 | 0.752 | 0.743 | 0.524 | 0.467 | 0.720 | 0.682 | 0.707 | 0.718 | 0.699 | 0.758 | 0.514 | 0.768 |
| DistalPhalanxTW | 0.640 | 0.612 | 0.698 | 0.676 | 0.669 | 0.676 | 0.568 | 0.590 | 0.671 | 0.694 | 0.695 | 0.685 | 0.610 | 0.663 | 0.285 | 0.591 |
| ProximalPhalanxOutlineCorrect | 0.835 | 0.856 | 0.887 | 0.859 | 0.866 | 0.873 | 0.770 | 0.784 | 0.807 | 0.768 | 0.907 | 0.866 | 0.730 | 0.920 | 0.684 | 0.817 |
| FreezerRegularTrain | 0.986 | 0.982 | 0.986 | 0.956 | 0.991 | 0.989 | 0.922 | 0.899 | 0.987 | 0.760 | 0.997 | 0.973 | 0.906 | 0.998 | 0.500 | 0.946 |
| TwoLeadECG | 0.793 | 0.847 | 0.986 | 0.999 | 0.993 | 0.976 | 0.871 | 0.905 | 0.877 | 0.784 | 0.999 | 0.806 | 0.753 | 1.000 | 0.500 | 0.949 |
| GunPointMaleVersusFemale | 0.991 | 0.991 | 1.000 | 0.997 | 0.994 | 0.997 | 1.000 | 0.997 | 0.977 | 0.978 | 0.997 | 0.952 | 0.980 | 0.992 | 0.525 | 0.988 |
| Trace | 1.000 | 1.000 | 1.000 | 0.990 | 1.000 | 1.000 | 1.000 | 1.000 | 0.952 | 0.740 | 1.000 | 0.902 | 0.806 | 1.000 | 0.240 | 0.934 |
| SmoothSubspace | 0.847 | 0.820 | 0.980 | 0.960 | 0.913 | 0.953 | 0.827 | 0.827 | 0.976 | 0.964 | 0.975 | 0.963 | 0.980 | 0.980 | 0.333 | 0.849 |
| MiddlePhalanxTW | 0.545 | 0.532 | 0.584 | 0.591 | 0.571 | 0.610 | 0.506 | 0.506 | 0.551 | 0.597 | 0.501 | 0.562 | 0.536 | 0.495 | 0.286 | 0.569 |
| SyntheticControl | 0.980 | 0.990 | 0.997 | 0.987 | 1.000 | 0.990 | 0.490 | 0.993 | 0.987 | 0.973 | 0.989 | 0.953 | 0.973 | 0.997 | 0.167 | 0.879 |
| ShapesAll | 0.807 | 0.815 | 0.902 | 0.848 | 0.788 | 0.773 | 0.733 | 0.768 | 0.617 | 0.679 | 0.894 | 0.599 | 0.776 | 0.926 | 0.017 | 0.643 |
| AllGestureWiimoteX | 0.569 | 0.607 | 0.777 | 0.763 | 0.703 | 0.697 | 0.259 | 0.716 | 0.411 | 0.475 | 0.713 | 0.261 | 0.477 | 0.741 | 0.100 | 0.522 |
| Wafer | 0.993 | 0.997 | 0.998 | 0.992 | 0.994 | 0.994 | 0.991 | 0.980 | 0.961 | 0.998 | 0.997 | 0.992 | 0.996 | 0.998 | 0.892 | 0.916 |
| FaceFour | 0.830 | 0.852 | 0.932 | 0.920 | 0.659 | 0.773 | 0.511 | 0.830 | 0.905 | 0.852 | 0.930 | 0.711 | 0.836 | 0.955 | 0.295 | 0.857 |
| CricketX | 0.721 | 0.749 | 0.782 | 0.713 | 0.623 | 0.731 | 0.385 | 0.754 | 0.535 | 0.644 | 0.794 | 0.513 | 0.591 | 0.799 | 0.074 | 0.627 |
| DistalPhalanxOutlineCorrect | 0.714 | 0.717 | 0.761 | 0.775 | 0.754 | 0.754 | 0.728 | 0.717 | 0.772 | 0.724 | 0.760 | 0.759 | 0.727 | 0.770 | 0.583 | 0.711 |
| ChlorineConcentration | 0.771 | 0.765 | 0.832 | 0.749 | 0.760 | 0.753 | 0.562 | 0.648 | 0.608 | 0.583 | 0.817 | 0.662 | 0.800 | 0.853 | 0.533 | 0.554 |
| Chinatown | 0.962 | 0.965 | 0.965 | 0.951 | 0.977 | 0.983 | 0.936 | 0.957 | 0.977 | 0.966 | 0.980 | 0.945 | 0.872 | 0.978 | 0.726 | 0.825 |
| GestureMidAirD1 | 0.654 | 0.646 | 0.608 | 0.608 | 0.431 | 0.369 | 0.208 | 0.569 | 0.534 | 0.528 | 0.695 | 0.518 | 0.575 | 0.698 | 0.038 | 0.549 |
| MiddlePhalanxOutlineAgeGroup | 0.494 | 0.461 | 0.636 | 0.656 | 0.643 | 0.630 | 0.617 | 0.500 | 0.534 | 0.577 | 0.535 | 0.558 | 0.522 | 0.545 | 0.571 | 0.578 |
| UMD | 0.951 | 0.993 | 1.000 | 0.993 | 0.993 | 0.986 | 0.910 | 0.993 | 0.960 | 0.771 | 0.988 | 0.842 | 0.949 | 0.990 | 0.333 | 0.835 |
| Crop | 0.733 | 0.734 | 0.756 | 0.722 | 0.738 | 0.742 | 0.710 | 0.665 | 0.670 | 0.760 | 0.738 | 0.687 | 0.618 | 0.743 | 0.042 | 0.489 |
| GesturePebbleZ1 | 0.808 | 0.849 | 0.930 | 0.919 | 0.378 | 0.395 | 0.500 | 0.791 | 0.844 | 0.821 | 0.880 | 0.769 | 0.792 | 0.901 | 0.163 | 0.840 |
| WordSynonyms | 0.668 | 0.688 | 0.676 | 0.691 | 0.630 | 0.531 | 0.422 | 0.649 | 0.568 | 0.557 | 0.561 | 0.470 | 0.599 | 0.617 | 0.219 | 0.506 |
| ArrowHead | 0.771 | 0.743 | 0.857 | 0.766 | 0.703 | 0.737 | 0.771 | 0.703 | 0.717 | 0.630 | 0.843 | 0.678 | 0.784 | 0.838 | 0.303 | 0.689 |
| Wine | 0.667 | 0.537 | 0.870 | 0.815 | 0.759 | 0.778 | 0.500 | 0.574 | 0.519 | 0.556 | 0.611 | 0.500 | 0.541 | 0.722 | 0.500 | 0.744 |
| Coffee | 0.929 | 0.893 | 1.000 | 1.000 | 1.000 | 1.000 | 0.821 | 1.000 | 1.000 | 0.886 | 1.000 | 0.979 | 0.993 | 1.000 | 0.507 | 0.979 |
| Earthquakes | 0.748 | 0.748 | 0.748 | 0.748 | 0.748 | 0.748 | 0.748 | 0.719 | 0.709 | 0.740 | 0.725 | 0.748 | 0.727 | 0.712 | 0.748 | 0.748 |
| Herring | 0.594 | 0.594 | 0.641 | 0.594 | 0.594 | 0.594 | 0.594 | 0.531 | 0.531 | 0.512 | 0.644 | 0.572 | 0.491 | 0.600 | 0.594 | 0.625 |
| Beef | 0.733 | 0.833 | 0.767 | 0.667 | 0.733 | 0.600 | 0.500 | 0.633 | 0.767 | 0.707 | 0.680 | 0.507 | 0.713 | 0.753 | 0.200 | 0.527 |
| MiddlePhalanxOutlineCorrect | 0.498 | 0.467 | 0.838 | 0.825 | 0.818 | 0.818 | 0.753 | 0.698 | 0.744 | 0.752 | 0.795 | 0.796 | 0.755 | 0.826 | 0.570 | 0.743 |
| ECGFiveDays | 0.740 | 0.804 | 1.000 | 1.000 | 0.999 | 0.878 | 0.763 | 0.768 | 0.874 | 0.842 | 0.985 | 0.800 | 0.973 | 0.966 | 0.497 | 0.723 |
| Yoga | 0.822 | 0.834 | 0.887 | 0.837 | 0.812 | 0.791 | 0.830 | 0.837 | 0.786 | 0.753 | 0.837 | 0.741 | 0.856 | 0.867 | 0.536 | 0.626 |
| Adiac | 0.629 | 0.688 | 0.762 | 0.675 | 0.726 | 0.767 | 0.550 | 0.604 | 0.393 | 0.318 | 0.841 | 0.620 | 0.391 | 0.833 | 0.023 | 0.428 |
| MoteStrain | 0.732 | 0.774 | 0.861 | 0.851 | 0.825 | 0.843 | 0.768 | 0.835 | 0.885 | 0.872 | 0.936 | 0.691 | 0.855 | 0.924 | 0.539 | 0.809 |
| Strawberry | 0.946 | 0.951 | 0.962 | 0.954 | 0.951 | 0.965 | 0.916 | 0.941 | 0.952 | 0.959 | 0.975 | 0.958 | 0.959 | 0.980 | 0.643 | 0.911 |
| InsectWingbeatSound | 0.609 | 0.607 | 0.630 | 0.597 | 0.549 | 0.415 | 0.266 | 0.355 | 0.585 | 0.630 | 0.392 | 0.587 | 0.604 | 0.499 | 0.091 | 0.435 |
| DodgerLoopWeekend | 0.826 | 0.826 | 0.964 | NaN | NaN | NaN | 0.732 | 0.949 | 0.974 | 0.983 | 0.904 | 0.978 | 0.978 | 0.952 | 0.739 | 0.954 |
| Meat | 0.933 | 0.917 | 0.950 | 0.950 | 0.917 | 0.883 | 0.900 | 0.933 | 0.913 | 0.787 | 0.803 | 0.787 | 0.893 | 0.990 | 0.333 | 0.970 |
| MelbournePedestrian | 0.886 | 0.876 | 0.959 | 0.944 | 0.942 | 0.949 | 0.741 | 0.791 | 0.813 | 0.884 | 0.912 | 0.840 | 0.863 | 0.909 | 0.100 | 0.730 |
| FaceAll | 0.760 | 0.791 | 0.771 | 0.786 | 0.766 | 0.813 | 0.504 | 0.808 | 0.774 | 0.794 | 0.938 | 0.720 | 0.794 | 0.867 | 0.080 | 0.673 |
| FacesUCR | 0.835 | 0.811 | 0.924 | 0.884 | 0.789 | 0.863 | 0.543 | 0.905 | 0.873 | 0.867 | 0.943 | 0.775 | 0.831 | 0.954 | 0.143 | 0.641 |
| AllGestureWiimoteY | 0.584 | 0.666 | 0.793 | 0.726 | 0.699 | 0.741 | 0.423 | 0.729 | 0.479 | 0.509 | 0.784 | 0.420 | 0.571 | 0.794 | 0.100 | 0.600 |
| ShakeGestureWiimoteZ | 0.840 | 0.960 | 0.940 | 0.920 | 0.820 | 0.860 | 0.760 | 0.860 | 0.580 | 0.756 | 0.884 | 0.516 | 0.548 | 0.880 | 0.100 | 0.864 |
| BME | 0.940 | 0.960 | 0.993 | 0.993 | 0.973 | 0.933 | 0.760 | 0.900 | 0.947 | 0.827 | 0.836 | 0.896 | 0.905 | 0.999 | 0.333 | 0.819 |
| FordB | 0.767 | 0.798 | 0.794 | 0.793 | 0.733 | 0.815 | 0.507 | 0.620 | 0.749 | 0.777 | 0.772 | 0.698 | 0.707 | 0.813 | 0.503 | 0.512 |
| Fish | 0.783 | 0.800 | 0.926 | 0.891 | 0.817 | 0.817 | 0.720 | 0.823 | 0.855 | 0.734 | 0.961 | 0.720 | 0.848 | 0.981 | 0.126 | 0.878 |
| SonyAIBORobotSurface2 | 0.827 | 0.829 | 0.871 | 0.889 | 0.834 | 0.907 | 0.745 | 0.831 | 0.831 | 0.844 | 0.980 | 0.804 | 0.831 | 0.975 | 0.617 | 0.635 |
| FiftyWords | 0.776 | 0.802 | 0.771 | 0.732 | 0.653 | 0.653 | 0.525 | 0.690 | 0.624 | 0.658 | 0.646 | 0.611 | 0.708 | 0.740 | 0.125 | 0.518 |
| ToeSegmentation1 | 0.912 | 0.925 | 0.917 | 0.939 | 0.864 | 0.930 | 0.807 | 0.772 | 0.598 | 0.706 | 0.961 | 0.559 | 0.589 | 0.957 | 0.526 | 0.882 |
| FreezerSmallTrain | 0.865 | 0.902 | 0.870 | 0.933 | 0.982 | 0.979 | 0.920 | 0.753 | 0.739 | 0.676 | 0.683 | 0.688 | 0.686 | 0.832 | 0.500 | 0.917 |
| TwoPatterns | 0.989 | 0.994 | 1.000 | 0.999 | 1.000 | 0.999 | 0.466 | 1.000 | 0.991 | 1.000 | 0.870 | 0.976 | 0.948 | 1.000 | 0.259 | 0.875 |
| ShapeletSim | 0.933 | 0.961 | 1.000 | 0.672 | 0.589 | 0.683 | 0.489 | 0.650 | 0.497 | 0.510 | 0.706 | 0.498 | 0.513 | 0.782 | 0.500 | 0.546 |
| Plane | 0.990 | 0.990 | 1.000 | 0.990 | 1.000 | 1.000 | 0.933 | 1.000 | 0.962 | 0.964 | 1.000 | 0.952 | 0.977 | 1.000 | 0.143 | 1.000 |
| GestureMidAirD3 | 0.300 | 0.369 | 0.292 | 0.285 | 0.292 | 0.177 | 0.154 | 0.323 | 0.317 | 0.368 | 0.326 | 0.278 | 0.382 | 0.340 | 0.038 | 0.275 |
| DiatomSizeReduction | 0.889 | 0.879 | 0.984 | 0.984 | 0.993 | 0.977 | 0.961 | 0.967 | 0.954 | 0.880 | 0.346 | 0.646 | 0.909 | 0.301 | 0.301 | 0.914 |
| CricketZ | 0.713 | 0.731 | 0.792 | 0.708 | 0.682 | 0.713 | 0.403 | 0.754 | 0.501 | 0.651 | 0.810 | 0.484 | 0.629 | 0.809 | 0.062 | 0.643 |
| Lightning7 | 0.712 | 0.726 | 0.863 | 0.795 | 0.767 | 0.685 | 0.411 | 0.726 | 0.647 | 0.696 | 0.825 | 0.559 | 0.616 | 0.827 | 0.260 | 0.608 |
| UWaveGestureLibraryY | 0.738 | 0.738 | 0.719 | 0.710 | 0.697 | 0.641 | 0.348 | 0.634 | 0.626 | 0.676 | 0.642 | 0.639 | 0.699 | 0.666 | 0.121 | 0.497 |
| GunPointAgeSpan | 0.959 | 0.962 | 0.987 | 0.994 | 0.984 | 0.994 | 0.991 | 0.918 | 0.912 | 0.890 | 0.996 | 0.887 | 0.934 | 0.997 | 0.494 | 0.965 |
| DistalPhalanxOutlineAgeGroup | 0.655 | 0.669 | 0.727 | 0.727 | 0.741 | 0.755 | 0.741 | 0.770 | 0.758 | 0.761 | 0.718 | 0.729 | 0.647 | 0.718 | 0.433 | 0.705 |
| SwedishLeaf | 0.914 | 0.923 | 0.941 | 0.914 | 0.880 | 0.923 | 0.738 | 0.792 | 0.884 | 0.902 | 0.967 | 0.841 | 0.845 | 0.963 | 0.064 | 0.837 |
| CBF | 0.972 | 0.960 | 1.000 | 0.983 | 0.983 | 0.998 | 0.898 | 0.997 | 0.959 | 0.977 | 0.994 | 0.908 | 0.869 | 0.996 | 0.332 | 0.896 |
| BeetleFly | 0.650 | 0.900 | 0.900 | 0.800 | 0.850 | 0.800 | 1.000 | 0.700 | 0.900 | 0.620 | 0.910 | 0.630 | 0.880 | 0.850 | 0.500 | 0.790 |
| AllGestureWiimoteZ | 0.504 | 0.537 | 0.746 | 0.723 | 0.646 | 0.689 | 0.447 | 0.643 | 0.375 | 0.396 | 0.692 | 0.287 | 0.439 | 0.726 | 0.100 | 0.516 |
| DodgerLoopDay | 0.475 | 0.438 | 0.562 | NaN | NaN | NaN | 0.200 | 0.500 | 0.312 | 0.487 | 0.143 | 0.305 | 0.160 | 0.150 | 0.160 | 0.593 |
| GunPointOldVersusYoung | 0.943 | 0.981 | 1.000 | 1.000 | 1.000 | 1.000 | 1.000 | 0.838 | 0.922 | 0.923 | 0.989 | 0.926 | 0.941 | 0.989 | 0.524 | 0.975 |
| FordA | 0.905 | 0.936 | 0.936 | 0.928 | 0.902 | 0.930 | 0.568 | 0.555 | 0.896 | 0.928 | 0.914 | 0.863 | 0.816 | 0.937 | 0.510 | 0.555 |
| ItalyPowerDemand | 0.941 | 0.911 | 0.925 | 0.954 | 0.928 | 0.955 | 0.845 | 0.950 | 0.954 | 0.964 | 0.963 | 0.966 | 0.953 | 0.962 | 0.499 | 0.871 |
| ProximalPhalanxOutlineAgeGroup | 0.873 | 0.863 | 0.834 | 0.844 | 0.854 | 0.839 | 0.854 | 0.805 | 0.812 | 0.872 | 0.825 | 0.839 | 0.849 | 0.847 | 0.488 | 0.839 |
| GunPoint | 0.927 | 0.927 | 0.980 | 0.980 | 0.967 | 0.993 | 0.827 | 0.907 | 0.948 | 0.784 | 1.000 | 0.907 | 0.928 | 0.991 | 0.493 | 0.989 |
| ProximalPhalanxTW | 0.732 | 0.712 | 0.824 | 0.771 | 0.810 | 0.800 | 0.780 | 0.761 | 0.777 | 0.791 | 0.761 | 0.775 | 0.767 | 0.773 | 0.341 | 0.784 |
| PickupGestureWiimoteZ | 0.600 | 0.620 | 0.820 | 0.740 | 0.620 | 0.600 | 0.240 | 0.660 | 0.608 | 0.496 | 0.744 | 0.412 | 0.604 | 0.704 | 0.100 | 0.616 |
| SonyAIBORobotSurface1 | 0.739 | 0.729 | 0.903 | 0.902 | 0.804 | 0.899 | 0.724 | 0.725 | 0.690 | 0.729 | 0.958 | 0.655 | 0.692 | 0.961 | 0.429 | 0.725 |
| PowerCons | 0.933 | 0.894 | 0.961 | 0.900 | 0.933 | 0.961 | 0.911 | 0.878 | 0.960 | 0.971 | 0.863 | 0.929 | 0.977 | 0.879 | 0.500 | 0.852 |
| PhalangesOutlinesCorrect | 0.686 | 0.652 | 0.809 | 0.784 | 0.787 | 0.804 | 0.773 | 0.728 | 0.799 | 0.745 | 0.818 | 0.795 | 0.756 | 0.845 | 0.613 | 0.656 |
| BirdChicken | 0.750 | 0.850 | 0.800 | 0.850 | 0.750 | 0.650 | 0.650 | 0.750 | 0.710 | 0.510 | 0.940 | 0.540 | 0.740 | 0.880 | 0.500 | 0.620 |
| ToeSegmentation2 | 0.915 | 0.915 | 0.892 | 0.900 | 0.831 | 0.877 | 0.615 | 0.838 | 0.752 | 0.702 | 0.889 | 0.649 | 0.745 | 0.894 | 0.815 | 0.794 |
| CricketY | 0.715 | 0.746 | 0.749 | 0.728 | 0.597 | 0.718 | 0.467 | 0.744 | 0.582 | 0.639 | 0.793 | 0.521 | 0.598 | 0.810 | 0.085 | 0.652 |
| ElectricDevices | 0.659 | 0.646 | 0.721 | 0.707 | 0.700 | 0.686 | 0.676 | 0.602 | 0.686 | 0.702 | 0.706 | 0.653 | 0.593 | 0.728 | 0.242 | 0.605 |
| DodgerLoopGame | 0.710 | 0.623 | 0.841 | NaN | NaN | NaN | 0.696 | 0.877 | 0.816 | 0.810 | 0.768 | 0.877 | 0.865 | 0.710 | 0.478 | 0.716 |
| Fungi | 0.828 | 0.898 | 0.957 | 1.000 | 0.527 | 0.753 | 0.366 | 0.839 | 0.961 | 0.934 | 0.018 | 0.051 | 0.863 | 0.177 | 0.063 | 0.439 |
| Symbols | 0.928 | 0.936 | 0.976 | 0.963 | 0.885 | 0.916 | 0.786 | 0.950 | 0.808 | 0.754 | 0.955 | 0.644 | 0.836 | 0.893 | 0.174 | 0.798 |
| UWaveGestureLibraryZ | 0.749 | 0.765 | 0.770 | 0.757 | 0.721 | 0.690 | 0.655 | 0.658 | 0.630 | 0.684 | 0.727 | 0.645 | 0.697 | 0.749 | 0.121 | 0.573 |
| ECG200 | 0.870 | 0.760 | 0.920 | 0.940 | 0.830 | 0.880 | 0.830 | 0.770 | 0.816 | 0.884 | 0.888 | 0.838 | 0.914 | 0.874 | 0.640 | 0.874 |

Table 11: **Classification accuracy** of methods across 91 UCR datasets. Results are shown for MOMENT$_0$ (without fine-tuning) and its pruned version, MOMENT$_{0\_pruned}$ (all blocks). We report that MOMENT$_0$ achieves 54 wins, 9 ties, and 28 losses compared to MOMENT$_{0\_pruned}$.

