# OpenReview forum: "Exploring Representations and Interventions in Time Series Foundation Models"
_ICML.cc/2025/Conference — ICML 2025 poster_

### Official Review · Reviewer_tr3o · 2025-02-28

**Overall Recommendation:** 2

**Summary:**

The paper delves into learned representations of time series foundation models. The authors evaluate the similarity of representations using CKA, revealing that larger models can learn redundant patterns. They propose a block-wise layer pruning strategy to reduce the feature dimensionality while keeping the performance. The authors also introduce a method to identify specific temporal patterns (e.g., trend and seasonality) from the latent space. They propose a steering method to guide the model to adjust predictions without fine-tuning.

**Claims And Evidence:**

The claims made in the paper are generally well-supported.

**Essential References Not Discussed:**

The paper adequately covers the relevant works, but it could benefit from a wider topic on time series foundation models, such as the scaling law of TSFMs.

**Experimental Designs Or Analyses:**

The authors conduct a detailed analysis of representation similarity, pruning effectiveness, and concept steering. However, the paper could benefit from additional experiments:

* The authors claim that: "Large TSFMs typically learn redundant representations, which often manifest as block-like structures in heatmaps”. Does this claim apply to non-pretrained time series models, e.g., a supervisedly trained Transformer on specific datasets? I wonder whether the mentioned redundant representations are caused by the variation redundancy of time series data.
* The use of synthetic data for concept identification and steering is a strong point. However, the paper could benefit from additional experiments on real-world datasets to further validate the generalizability of the steering method.

**Methods And Evaluation Criteria:**

The proposed methods and evaluation criteria (CKA and LDA) are appropriate for the researched problems.

**Other Comments Or Suggestions:**

Several charts in the experimental part are not very readable, such as the meaning of numbers in Figure 4 and Figure 5.

**Other Strengths And Weaknesses:**

Strength: The paper addresses a timely topic in the field of time series analysis, particularly the interpretability and controllability of large foundation models.

Weakness:

* The proposed pruning method may lack novelty. What is the main inovation beyond Nguyen's prior works?
* The inference time saved by the proposed block-wise pruning seems to be marginal (Table 7, 20.88ms -> 19.82ms). How does this approach compare to other pruning techniques in relevant literature?
* The motivation of the steering method is not well presented. What is the benefit (or applications) to guide the model for adjusted predictions with post-informed trend and seasonality?
* The visualization of the experiment may be derived from a subset of samples. It would be beneficial for the author to explain the rationale behind the selection of these samples and provide relevant statistics to mitigate the risk of cherry-picking.

**Questions For Authors:**

See above.

**Relation To Broader Scientific Literature:**

The authors build on prior work for representation analysis of vision models, extending these ideas to the domain of time series foundation models.

**Theoretical Claims:**

The paper does not make strong theoretical claims, focusing on empirical analysis.

---

> ### Author Rebuttal · Authors · 2025-04-01
>
> Dear Reviewer tr3o,
>
> Thank you for your thoughtful and detailed review of our paper. We appreciate that you found our "analysis of representation similarity, pruning effectiveness, and concept steering to be detailed and that the claims made in the paper are generally well-supported."
>
> We have addressed your concerns and questions below, and we'd be happy to address any further questions or comments. *If you feel that we have addressed your concerns, we respectfully ask that you consider increasing your score.*
>
> **TSFM representational redundancy**: This is a great point! We found that both supervised models trained on specific datasets, and pretrained text & vision models exhibit representational similarity. While this is common knowledge for other models, prior to our work, we were unsure if this holds for TSFMs. Just like prior work, we provide no explanation behind this phenomenon, but we agree that redundancy in the training data may cause this phenomenon. More importantly, we leverage this finding to aggressively prune TSFMs, and make their use more pragmatic in real-world tasks.
>
> **Additional experiments on real-world datasets:** Please refer to section **Cherry-picked example from Steering** in Rebuttal to Lqam.
>
> ### Questions
> 1. **Novelty of Proposed Pruning Method** We assume that you are referring to this paper in the following paper [1]. Let us know if we got the wrong paper. While our approach is inspired by Nguyen et al. (2021), it differs in key ways: (1) The authors remove individual layers, whereas we prune entire self-similar blocks at once, preserving their boundaries to maintain a notion of continuity in the model’s internal representations. (2) We are also one of the first to apply this type simple block-level pruning to large pretrained Time Series Foundation Models. (3) Finally, we validate our approach on real-world tasks beyond classification, such as imputation and forecasting-demonstrating up to 52% inference speedup with minimal performance degradation, even in zero-shot settings.
>
> [1] Nguyen, T., Raghu, M., and Kornblith, S. Do Wide and Deep Networks Learn the Same Things? Uncovering How Neural Network Representations Vary with Width and Depth. In International Conference on Learning Representations, 2021.
>
> 2. **Inference Time of Pruning:** You're absolutely right that the inference time improvements reported in Table 7 were marginal—this was due to a bug in our original implementation. Initially, pruning was done by zeroing out weights, which left computation intact and yielded minimal speedups, without utilizing CUDA kernels for sparse computations. After correcting this and implementing a computational graph-level pruning mechanism that fully skips computation of pruned blocks, we observed substantial speedups, with inference time reduced by up to 52% (e.g., from 23.17 ms to 11.16 ms for MOMENT, and from 31.68 ms to 15.33 ms for Chronos). Compared to prior pruning techniques that focus on sparsifying weights or pruning individual layers, our method is simple, structurally aligned with learned representations. These updated results are included in the paper.
>
> 3. **Motivation of Steering:** The motivation behind concept steering is to enable controlled, post-training updates to model embeddings, allowing models to incorporate new concepts or events into predictions without requiring additional training or fine-tuning. This approach offers several practical benefits: (1) It enables users to imbue models with new or missing contextual factors after training, which is particularly valuable when models weren't originally trained on certain scenarios. For example, we can steer vital signs in healthcare based on new treatments or adjust financial forecasts based on emerging events such as positive earnings surprises. Adding these contextual factors, even as simple trends, has significant implications for improving model predictions in zero-shot and out-of-distribution scenarios. (2) Steering also supports synthetic data generation of realistic time series variations. In our experiments with the ECG Arrhythmia classification dataset (ECG5000), we demonstrate that time series classified as normal heartbeats can be steered to produce time series classified as abnormal heartbeats. Such data generation capabilities can be used to augment data for model training or generate new samples for us to better understand the decision boundaries in model predictions.
>
> 4. **Risk of cherry-picking:** Please refer to section **Cherry-picked example from Steering** in Rebuttal to Lqam.

---

### Official Review · Reviewer_Lqam · 2025-03-11

**Overall Recommendation:** 3

**Summary:**

The paper performs analyses into 3 time series foundation models, Chronos, Moirai, and MOMENT. Using concepts from the interpretability literature, the paper studies i) representation similarity across layers, ii) identification of human interpretable concepts, and iii) model intervention. Via experiments, the paper shows that these models have significant similarity between layers, which can ultimately be pruned and still retain performance. They show that these models indeed learn human interpretable concepts such as (linear) trend and seasonality, and the models can be manipulated into making predictions with these concepts.

**Claims And Evidence:**

The paper is largely an experimental analysis into existing models, and very nicely sets up the experimental design to support their claims. Evidence regarding representational similarity is clear and convincing. However the following 2 points are problematic:

1. "Block-wise pruning can improve model throughput, without compromising accuracy." seems to be an overstatement given that the subsequent evidence provided showed minimal throughput improvement (see questions section), and zero-shot performance is not retained. I recommend the authors to reduce the boldness of the statement.

2. The evidence provided for the "concepts" and "interventions" parts, specifically figures 7 and 8 are unclear. I am unable to understand what the figures mean. For figure 7, it is unclear what is the relationship between the red line and the heat map are, and it is also unclear what the heat map is trying to convey. For figure 8, it took me some time to understand the differences between steering vs compositional steering - the sentences "Introduce periodicity (i) and trend (ii, iii) to constant time series" and "Introduce trend and periodicity to constant time series" should make it clearer the difference between steering vs compositional steering. I would further recommend "(MOMENT (top), Chronos (bottom))" to be labels at the side of the diagram instead. I do not understand what the lines represent, are they ground truth time series? or model predictions? What is the difference between the dark lines vs transparent lines? Where is the demarcation between Chronos inputs vs forecasts?

**Essential References Not Discussed:**

None.

**Experimental Designs Or Analyses:**

The experimental design is largely sound, my main concern is with regards to the steering section, which uses a single example which could be cherry picked. It would be better to have multiple of such plots, or some kind of dataset level experiment to show this capability. Figure 10 looks promising, but it is unclear how the dataset in appendix A has been used for this set of experiments. More explanation should be given.

**Methods And Evaluation Criteria:**

The methods and evaluations make sense for the analyses. The paper clearly lays out research questions that it intends to explore, and presents the tools and methods it uses to investigate the questions. The paper is comprehensive in exploring 3 different models.

**Other Comments Or Suggestions:**

### Nits

1. CKA should be defined at the first usage in line 46, right column, or avoid the use of the term CKA in that part of the introduction, just mention "similarity" without mentioning the metric used.

2. Line 216, right column - ...  identify which layer l in ... - "l" is not really required as it is not referred to again.

3. Line 225, left column - the term "residual stream" should receive a brief explanation and citation as it is not standard Transformer parlance, but more of a term used within the mech interp community.

### Suggestions

4. Line 291, left column - Fig. 7 should be fig. 5?

5. Fig 4 - Label x and y axes, especially for tiny vs mini

6. Table 4 - include percentage change

7. Line 304, right column - "... inference speed compared to unpruned ..." - compared -> comparable

**Other Strengths And Weaknesses:**

1. Writing structure can be improved for easier reading. The writing structure which lists all methodology first, followed by experimental results does not suit this paper. Instead, this paper should take the approach of having the experimental results right after each method subsection, e.g. Explain representational similarity -> similarity results -> explain pruning -> pruning results -> explain time series concepts -> time series concepts results, and so on.

2. Regarding the pruning algorithm, the key idea should be on how to detect blocks rather than how to prune them. While an automatic algorithm to detect them is presented in the appendix, ultimately the blocks for experiments were identified visually (as stated in Appendix C.2).

3. The biggest weakness of this paper is its clarity. Starting from the definition $h_i^{(j)} \in \mathbb{R}^{n \times D}$, it is unclear what $n$ is. It seems that indices $I, j$ can be dropped, since all variables are indexed by them. "Additionally, we perform probing on representations averaged along the token dimension for each i-th layer." - not sure what this statement means. Formal notation should be used to define $\mu_s, \mu_c, \sigma_s, \sigma_c$. It's not clear why a "mean embedding value" is a scalar. Clarity of experimental results are also an issue which have been mentioned above.

**Questions For Authors:**

1. Line 226, left column - $h_i^{(j)} \in \mathbb{R}^{n \times D}$ - what is $n$?

2. How was pruning actually performed, especially in the experiments? Algorithm 1 says "zero out the weights". Was inferencing skipped on those layers or not? How would zero-ing out the weights but not skipping the layers incur a speed up?

3. Table 7 - I don't understand where the improvement in inference speed comes from (related to previous question). If it is the case that the layers were skipped, then I'm confused why the improvement is so small, i.e. only 1 ms for MOMENT, and the relationship between block 1/2/3 vs all blocks doesn't make sense to me.

**Relation To Broader Scientific Literature:**

The paper raises some interesting findings for time series foundation models, and suggests several avenues for deeper exploration. Firstly, it raises the issue of redundancy amongst the layers and suggests that existing models can be much more parameter efficient. It also shows that these models can be steered based on a set of examples - this can possibly be extended to domain based steering.

**Theoretical Claims:**

No theoretical claims made.

---

> ### Author Rebuttal · Authors · 2025-04-01
>
> Dear Reviewer Lqam,
>
> Thank you so much for your time! We are glad that you found that our paper "very nicely sets up the experimental design to support their [our] claims, is comprehensive in exploring 3 different models, and presents clear and convincing evidence regarding representational similarity". We have addressed all your comments and suggestions in the current version of the manuscript.
>
> Below, we address your concerns and questions. If you have any further comments, we'd be happy to address them. *If you feel that we have addressed your concerns, we respectfully ask that you consider increasing your score.*
>
> **Cherry-picked example from Steering**: We appreciate the reviewer’s suggestion to include more plots to showcase the effect of steering. To demonstrate that the steering effect is not limited to a single time series example, we generated datasets with different random seeds and assessed that the same phenomena of linear separability happens, with successful steering from one group to another that is visible in the visualizations. Also, visualizations that show influence of a specific steering intervention in the output of the model also are repeatable across different samples, datasets and setups.
>
> We also provided additional steering results with real-world data to showcase its effectiveness for other datasets in Appendix F.1. Using our proposed steering approach in MOMENT with time series from a real-world ECG Arrhythmia classification dataset (ECG5000), we demonstrate that time series classified as normal heartbeats can be steered to time series classified as abnormal heartbeats. This result confirms that concept steering can effectively guide time series into different clinically relevant category classifications, which could help enhance our understanding of time series pattern variations that guide model predictions for medical condition classification. Additionally, such steering results highlight the potential utility of the method for synthetic data generation. To provide further evidence of the effects of steering across datasets in different domains, we are currently running experiments on steering concepts in popular forecasting (ETT-h1), classification (MIT-BIH ECG dataset), and anomaly detection (UCR Anomaly Archive) experiments.
>
> **Unclear Figures 7 and 8**: We agree that figures 7 & 8 can be made more clear. We have made the following edits to improve clarity.
>
> Figure 7- The purpose of these figures is to show where in the model linear separability peaks (as measured by normalized Fisher’s LDA objective range (0,1)). X axis is the model depth - in terms of layers, Y axis is the position of the patch of time series from which we took embeddings. The more yellow the specific (layer, patch) combination the more drastic the separation was. The red line also showcases the linear separability - the idea here was that we averaged embeddings across the patch dimension, so that we have a single time series embedding at the specific layer. Here, the X axis is also model depth, while Y axis is the degree of linear separability. In addition to updating the figure caption with this information, we have clarified that the heatmap and red line use distinct y-axes, which was not immediately clear in the original figure. To improve readability, we have updated the plot by adding labeled axes: the x-axis is now labeled “Model Depth,” the left y-axis (for the heatmap) is labeled “Patch Position,” and the right y-axis (for the red line) is labeled “Fisher’s LDA.”
>
> Figure 8 - we’ve incorporated the suggestions about improving the captions. To clarify the setup-we provided constant time series as an input to the model (not visualized, information provided in the caption), and visualize model outputs with and without steering applied, which are referred to as ‘Perturbed’ and ‘Non-perturbed’ outputs in the figure legend, respectively. As expected, without steering applied, the model outputs a constant time series signal. With steering applied, we obtain a different intended concept output, reflecting a trend, seasonality, or a combination of both, depending on the Beta parameter. For each model output, we show the raw output (lighter color) and the moving average (darker color), which helps filter out noise that is an artifact of the model. We have updated the figure caption to include this information.
>
> ### Questions
> 1. **Line 226 – What is n?** The symbol $n$ refers to the number of time series samples in the dataset. Each hidden representation $h_i^{(j)}$ corresponds to the output of the $j$-th layer at $i$-th token for all $n$ samples considered, with each embedding having dimensionality $D$.
>
> 2. **On Blockwise Pruning and Table 7:** Thank you for your thoughtful observations. Please refer to section **Inference Time of Pruning** in our rebuttal to Reviewer tr30.

---

### Official Review · Reviewer_i5c5 · 2025-03-14

**Overall Recommendation:** 3

**Summary:**

This paper investigates the internal workings of time series foundation models by analyzing their learned representations. It reveals that these models exhibit block-like redundancy across layers, which can be exploited through block-wise pruning to reduce model size and improve inference speed without compromising accuracy. The study further identifies human-interpretable concepts—such as trends, periodicity, and seasonality—within the latent space and demonstrates how interventions in this space (concept steering) can guide model predictions toward desired outcomes. Overall, the work provides valuable insights for optimizing and controlling time series forecasting models.

**Claims And Evidence:**

yes, they are.

**Essential References Not Discussed:**

All good.

**Experimental Designs Or Analyses:**

Yes, I checked all. There are no obvious issues.

**Methods And Evaluation Criteria:**

Yes, they do.

**Other Comments Or Suggestions:**

None.

**Other Strengths And Weaknesses:**

All have been discussed.

**Questions For Authors:**

None.

**Relation To Broader Scientific Literature:**

It extends representation similarity analysis—originally developed in the computer vision and NLP communities.

**Theoretical Claims:**

There are no theoretical claims

---

> ### Author Rebuttal · Authors · 2025-04-01
>
> Dear Reviewer i5c5,
>
> Thank you for reviewing our paper. We appreciate your recognition of our work's contributions regarding model redundancy patterns, block-wise pruning, and interpretable latent space concepts.
>
> Given your "Weak accept" recommendation, we'd be grateful if you could let us know how we might improve our paper to strengthen your assessment. Are there specific aspects of our analysis, experimental design, or presentation that we could improve to better convey the significance and impact of our contributions?
>
> Thank you again for your time and consideration.
>
> Sincerely,
>
> The Authors

---

### Decision · Program_Chairs · 2025-05-01

**Decision:**

Accept (poster)

**Comment:**

This paper studies the representation of time series foundation models. By using representation similarity across multiple models, they identify redundant layer-wise representations, subsequently developing block-wise pruning schemes to improve runtime while maintaining performance. The reviewers had a number of comments, mostly with the clarity of the approach and exposition, which included several major issues including errors in computed numbers that were later fixed, and lack of clarity on novelty over a similar related work. Regardless the contributions to examining foundation models were appreciated and so if there is room I would recommend acceptance.